# $\mathrm{STORM}^+$: Fully Adaptive SGD with Momentum for Nonconvex Optimization

**Kfir Y. Levy** [*]
Technion
kfirylevy@technion.ac.il

**Ali Kavis**
EPFL
ali.kavis@epfl.ch

**Volkan Cevher**
EPFL
volkan.cevher@epfl.ch

## Abstract

In this work we investigate stochastic non-convex optimization problems where the objective is an expectation over smooth loss functions, and the goal is to find an approximate stationary point. The most popular approach to handling such problems is variance reduction techniques, which are also known to obtain tight convergence rates, matching the lower bounds in this case. Nevertheless, these techniques require a careful maintenance of anchor points in conjunction with appropriately selected "mega-batchsizes". This leads to a challenging hyperparameter tuning problem, that weakens their practicality. Recently, [Cutkosky and Orabona, 2019] have shown that one can employ recursive momentum in order to avoid the use of anchor points and large batchsizes, and still obtain the optimal rate for this setting. Yet, their method called STORM crucially relies on the knowledge of the smoothness, as well a bound on the gradient norms. In this work we propose $\mathrm{STORM}^+$, a new method that is completely parameter-free, does not require large batch-sizes, and obtains the optimal $O(1/T^{1/3})$ rate for finding an approximate stationary point. Our work builds on the STORM algorithm, in conjunction with a novel approach to adaptively set the learning rate and momentum parameters.

## 1 Introduction

Over the past decade non-convex models have become principal tools in ML (Machine Learning), and in data-science. This predominantly includes deep models, as well as Phase Retrieval [Candes et al., 2015], non-negative matrix factorization [Hoyer, 2004], and matrix completion problems [Ge et al., 2016] among others.

The main workhorse for training ML models is SGD (stochastic gradient descent) and its numerous variants. One parameter that significantly affects the SGD performance is the learning rate, which often requires a careful and costly hyper-parameter tuning. Adaptive approaches to setting the learning rate like AdaGrad [Duchi et al., 2011] and Adam [Kingma and Ba, 2014] as well as non-adaptive heuristics [Loshchilov and Hutter, 2017, He et al., 2019] are very popular in modern ML applications, yet these methods also require some tuning of hyper-parameters like momentum and the scale of the learning rate schedule.

A popular SGD heuristic that has proven to be crucial in many applications is the use of *momentum*, i.e., the use of a weighted average of past gradients instead of the current gradient [Sutskever et al., 2013, Kingma and Ba, 2014]. Although adaptive approaches to setting the momentum have been investigated in the past [Srinivasan et al., 2018, Hameed et al., 2016], principled and theoretically-grounded approaches to doing so are less investigated. Another aspect that has not been extensively studied, which we take into account in this work, is the interplay between learning rate and momentum.

---

[*]A Viterbi fellow. Corresponding author

35th Conference on Neural Information Processing Systems (NeurIPS 2021).

In this work we explore momentum-based adaptive and parameter-free methods for stochastic non-convex optimization problems. Concretely, we focus on the setting where the objective is an *expectation over smooth losses* (see Eq. (4)), and the goal is to find an approximate stationary point.

In the general case of smooth non-convex objectives it is known that one can approach a stationary point at a rate of $O(1/T^{1/4})$, where $T$ is the total number of samples [Ghadimi and Lan, 2013]. While this rate is optimal in the general case, it is known that one can obtain an improved rate of $O(1/T^{1/3})$ if the objective is an *expectation over smooth losses* [Fang et al., 2018, Zhou et al., 2018, Cutkosky and Orabona, 2019, Tran-Dinh et al., 2019]. Besides, this rate was recently shown to be tight [Arjevani et al., 2019].

Nevertheless, most of the methods developed for this setting rely on variance reduction techniques [Johnson and Zhang, 2013, Zhang et al., 2013, Mahdavi et al., 2013, Wang et al., 2013], which require careful maintenance of anchor points in conjunction with appropriately selected large batchsizes. This leads to a challenging hyper-parameter tuning problem, weakening their practicality. One exception is the recent STORM algorithm of Cutkosky and Orabona [2019].

STORM does not require large batches nor anchor points; instead, it uses a corrected momentum based gradient update that leads to implicit variance reduction, which in turn facilitates fast convergence. Unfortunately, none of the aforementioned methods (including STORM ) is parameter-free. Indeed, the knowledge of smoothness parameter together with either the noise variance or a bound on the norm of the gradients are crucial to establish their guarantees.

In this work, we essentially develop a parameter-free variant of STORM algorithm. We summarize our contributions as follows,

- We present STORM$^+$ , a *parameter-free* momentum based method that ensures the optimal $O(1/T^{1/3})$ rate for the *expectation over smooth losses* setting. Similarly to STORM , our method does not require large-batches nor anchor points.

- STORM$^+$ implicitly adapts to the variance of the gradients. Concretely, it obtains convergence rate of $O(1/\sqrt{T} + \sigma^{1/3}/T^{1/3})$, which recovers the optimal $O(1/\sqrt{T})$ rate in the noiseless case. We also improve over STORM by shaving off a $(\log T)^{3/4}$ factor from the $1/\sqrt{T}$ term.

- In STORM$^+$ we demonstrate a novel way to set the learning rate by introducing an adaptive interplay between learning rate and momentum parameters.

## 2  Related Work

In the context of stochastic non-convex optimization with general smooth losses, it was shown in Ghadimi and Lan [2013] that SGD with an appropriately selected learning rate can obtain a rate of $O(1/T^{1/4})$ for finding an approximate stationary point, which is known to match the respective lower bound [Arjevani et al., 2019]. While the method of Ghadimi and Lan [2013] requires knowledge of the smoothness and variance parameters, recent works have shown that adaptive methods like AdaGrad are able to obtain this bound in a parameter free manner, as well as to adapt to the variance of the problem [Li and Orabona, 2019, Ward et al., 2019, Reddi et al., 2018]. These results, in a sense, explain the success of adaptive[2] methods like AdaGrad [Duchi et al., 2011], Adam [Kingma and Ba, 2014], and RMSProp [Tieleman and Hinton, 2012] in handling non-convex problems.

The idea of using variance reduction techniques for non-convex problems was first suggested in the context of finite sum problems by Allen-Zhu and Hazan [2016], Reddi et al. [2016], showing a rate of $O(1/T^{1/4})$. This was later improved by Lei et al. [2017] to a rate of $O(1/T^{3/10})$. The first works that have obtained the optimal $O(1/T^{1/3})$ for this setting were Fang et al. [2018], Zhou et al. [2018]. Additionally, Fang et al. [2018] shows that the same convergence behavior applies to the more general *expectation over smooth losses* setting (see Eq. (4)) – a setting that captures finite-sum problems as a private case.

---

[2]An adaptive method is a method that updates its learning rate according to the (noisy) gradient feedback that it receives throughout the training process.

The STORM algorithm suggested in Cutkosky and Orabona [2019] is the first algorithm to obtain the optimal $O(1/T^{1/3})$ for this setting without the need to maintain anchor points or large batches. Instead, it relies on a clever correction of the momentum by making only one extra call to the oracle, which leads to an implicit variance reduction effect. Moreover, STORM adapts to the variance of the problem by obtaining a rate of $O((\log T)^{3/4}/\sqrt{T} + \sigma^{1/3}/T^{1/3})$ without any prior knowledge of variance. However, it needs to know the smoothness parameter and a bound on the gradient norms to set the step size and momentum parameters. Simultaneously to the work of Cutkosky and Orabona [2019], another paper [Tran-Dinh et al., 2019] have obtained the same optimal bound by proposing a similar update rule. Note that Tran-Dinh et al. [2019] does calculate a single anchor point, and it still requires the knowledge of the smoothness and variance parameters.

## 3 Setting and Preliminaries

We discuss stochastic non-convex optimization problems where the objective $f : \mathbb{R}^d \mapsto \mathbb{R}$ is of the following form,
$$f(x) := \mathbf{E}_{\xi \sim \mathcal{D}}[f(x; \xi)] ,$$
and $\mathcal{D}$ is an unknown distribution from which we may draw i.i.d. samples. Our goal is to find an approximate stationary point of $f$, i.e. after $T$ draws from $\mathcal{D}$ we should output a point $\bar{x} \in \mathbb{R}^d$ such that $\mathbf{E}\|\nabla f(x)\| \le \text{Poly}(1/T)$.

We focus on first order methods, i.e., methods that may access the gradients of $f(\cdot, \xi)$, and make the following assumptions regarding the noisy gradients and function values.

**Bounded values:** There exists $B > 0$ such that,
$$\max_{x,y \in \mathbb{R}^d} |f(x) - f(y)| \le B. \tag{1}$$

**Bounded gradients:** There exists $G > 0$ such that,
$$\|\nabla f(x; \xi)\|^2 \le G^2 ; \quad \forall x \in \mathbb{R}^d, \xi \in \mathbf{support}\{\mathcal{D}\}. \tag{2}$$

**Bounded variance:** There exists $\sigma > 0$ such that,
$$\mathbf{E}\|\nabla f(x; \xi) - \nabla f(x)\|^2 \le \sigma^2 ; \quad \forall x \in \mathbb{R}^d. \tag{3}$$

**Expectation over smooth losses:** There exists $L > 0$ such that,
$$\|\nabla f(x; \xi) - \nabla f(y; \xi)\| \le L\|x - y\| ; \quad \forall x, y \in \mathbb{R}^d, \xi \in \mathbf{support}\{\mathcal{D}\} . \tag{4}$$

The last assumption also implies that the expected loss $f(\cdot)$ is $L$ smooth. A property of smooth functions that we will exploit throughout the paper is the following,
$$f(y) \le f(x) + \nabla f(x)^\top (y - x) + (L/2)\|y - x\|^2 ; \quad \forall x, y \in \mathbb{R}^d \tag{5}$$

In the rest of this manuscript, $\nabla f(x; \xi)$ relates to gradients with respect to $x$, i.e., $\nabla := \nabla_x$. We use $\|\cdot\|$ to denote the Euclidean norm, and $x^*$ denotes a global minima of $f(\cdot)$, i.e., $x^* = \min_{x \in \mathbb{R}^d} f(x)$.

## 4 Method

In this section we present $\text{STORM}^+$ (STochastic Recursive Momentum +): a parameter-free stochastic optimization method that finds approximate stationary points at an optimal rate. We describe our method in Alg. 1 and Eq. (8), and state its guarantees in Theorem 1.

**The original** STORM **algorithm:** The original STORM template of Cutkosky and Orabona [2019] relies on an SGD-style update with a corrected momentum. Concretely, the idea is to maintain a gradient estimate $d_t$ which is a *corrected* weighted average of past stochastic gradients, and then update the iterates similarly to SGD,
$$x_{t+1} = x_t - \eta_t d_t . \tag{6}$$

Standard momentum is a weighted average of past gradients,
$$d_t = a_t \nabla f(x_t, \xi_t) + (1 - a_t)d_{t-1} ; \quad \text{where } a_t \in [0, 1] .$$

Under this construction, $d_t$ is generally a biased estimate of $\nabla f(x_t)$. In STORM it is suggested to add a correction term ,$(1 - a_t)(\nabla f(x_t, \xi_t) - \nabla f(x_{t-1}, \xi_t))$, which leads to the following update rule (again, $a_t \in [0, 1]$),

$$d_t = \nabla f(x_t, \xi_t) + (1 - a_t)(d_{t-1} - \nabla f(x_{t-1}, \xi_t)) , \qquad \text{(Corrected Momentum)}$$

The correction term plays a crucial role here: it exploits the smoothness of $f(\cdot, \xi)$ in a way that leads to a variance reduction effect. To see this effect one can inspect the error of the momentum $d_t$ compared to the exact gradient at $x_t$,

$$\epsilon_t := d_t - \nabla f(x_t) .$$

The STORM update rule induces the following error dynamics,

$$\epsilon_t = (1 - a_t)\epsilon_{t-1} + a_t(\nabla f(x_t, \xi_t) - \nabla f(x_t)) + (1 - a_t)Z_t$$

where $Z_t := (\nabla f(x_t, \xi_t) - \nabla f(x_{t-1}, \xi_t)) - (\nabla f(x_t) - \nabla f(x_{t-1}))$. Due to the smoothness of the objective we have $\|Z_t\| \leq O(\|x_t - x_{t-1}\|) = O(\eta_{t-1}\|d_{t-1}\|)$. Intuitively, as we approach a stationary point (and use a small enough learning rate) then $\eta_{t-1}\|d_{t-1}\|$ decreases which in turn reduces the magnitude of $Z_t$'s. Moreover, the second term in the above dynamics, $a_t(\nabla f(x_t, \xi_t) - \nabla f(x_t))$, can be controlled by choosing a small enough momentum $a_t$. Thus, carefully controlling the learning rate and momentum parameters leads to a variance reduction effect which facilitates fast convergence.

The original STORM paper [Cutkosky and Orabona, 2019] makes the following choices,

$$\eta_t = \theta \Big/ \left( w + \sum_{i=1}^{t} \|g_i\|^2 \right)^{1/3} \qquad \& \qquad a_t = cL^2\eta_{t-1}^2 , \qquad (7)$$

where we denote $g_t := \nabla f(x_t, \xi_t)$. The above choice of learning rate is inspired by AdaGrad Duchi et al. [2011], which also sets the learning rate inversely proportional to the cumulative square norms of past gradients. Note that $\theta$ and $w$ are constants that depend on the smoothness of the objective $L$, as well as on the bound on the gradients $G$, and $c$ is an absolute constant independent of the problem's characteristics. These choices of the constants and especially the choice of $a_t \propto L^2\eta_{t-1}^2$ is crucial for the analysis of the original STORM . In fact, the convergence proof for STORM breaks down unless we encode this prior knowledge into $\eta_t$ and $a_t$. Next, we describe our parameter-free version.

**Our STORM$^+$ algorithm:** STORM$^+$ relies on the original STORM template described in Equations (6) and (Corrected Momentum), with the following parameter-free choices of learning rate and momentum parameter,

$$\eta_t = 1 \Big/ \left( \sum_{i=1}^{t} \|d_i\|^2 / a_{i+1} \right)^{1/3} \qquad \& \qquad a_t = 1 \Big/ \left( 1 + \sum_{i=1}^{t-1} \|g_i\|^2 \right)^{2/3} , \qquad (8)$$

where again we denote $g_t := \nabla f(x_t, \xi_t)$. Note that in contrast to the original STORM our adaptive learning rate builds on history of estimates $\{d_1, \ldots, d_t\}$ as well as on the momentum parameters $\{a_1, \ldots, a_{t+1}\}$. Our momentum term is similar to the adaptive choice of STORM , yet it does not require a bound on the gradients nor on the smoothness parameter, which was crucial for the original analysis. Finally, note that the above choice ensures $a_t \in [0, 1]$.

For completeness we present our method in Alg. 1, where it can be seen that STORM$^+$ is a combination of the original STORM template (Equations (6) and (Corrected Momentum)) together with the specific choices of $\eta_t$ and $a_t$ appearing in Eq. (8). Note that the solution that STORM$^+$ outputs is a point chosen uniformly at random among all iterates, which is quite standard in (stochastic) non-convex optimization.

*Notation:* In Alg. 1 and throughout the rest of the paper we will employ the following notation,

$$g_t := \nabla f(x_t, \xi_t) ; \quad \tilde{g}_t := \nabla f(x_t, \xi_{t+1}) ; \quad \bar{g}_t := \nabla f(x_t) .$$

Now, we are at a position to present our main theorem regarding STORM$^+$ (Alg. 1):

**Algorithm 1** STORM$^+$

---

**Input:** #iterations $T$, $x_1 \in \mathbb{R}^d$
1: Sample $\xi_1$ and set $d_1 = g_1 = \nabla f(x_1, \xi_1)$
2: **for** $t = 1, ..., T$ **do**
3: $\qquad a_{t+1} \leftarrow 1/\left(1 + \sum_{i=1}^{t} \|g_i\|^2\right)^{2/3} \quad \& \quad \eta_t \leftarrow 1/\left(\sum_{i=1}^{t} \|d_i\|^2/a_{i+1}\right)^{1/3}$
4: $\qquad x_{t+1} \leftarrow x_t - \eta_t d_t$
5: $\qquad$ Sample $\xi_{t+1}$ and set $g_{t+1} := \nabla f(x_{t+1}; \xi_{t+1})$, and $\tilde{g}_t := \nabla f(x_t; \xi_{t+1})$
6: $\qquad d_{t+1} \leftarrow g_{t+1} + (1 - a_{t+1})(d_t - \tilde{g}_t)$
7: **end for**
8: Choose $\bar{x}_T$ uniformly at random from $\{x_1, \ldots, x_T\}$
9: **return** $\bar{x}_T$

---

**Theorem 1.** *Under the assumption in Eq.* (1), (2), (3) *and* (4) *in Section* 3, STORM$^+$ *ensures,*

$$\mathbf{E}\|\nabla f(\bar{x}_T)\| \leq O\left(\frac{M}{\sqrt{T}} + \frac{\kappa\sigma^{1/3}}{T^{1/3}}\right),$$

*where* $\kappa = O(B^{3/4} + L^{3/2})$; $M = O(1 + L^{9/4} + B^{9/8} + G^5 + (LG^4)^{3/2})$, *and the expectation is with respect to the randomization of the samples as well as the algorithm's.*

Theorem 1 demonstrates that in the stochastic case STORM$^+$ achieves the optimal $O(1/T^{1/3})$ rate for our setting. Moreover, it can be seen that STORM$^+$ implicitly adapts to the variance of the noise; in the noiseless case where $\sigma = 0$, STORM$^+$ recovers the optimal $O(1/\sqrt{T})$ rate. We note that scaling the learning rate by some (absolute) constant factor may enable us to obtain better dependence on $L$ and $B$.

## 5 Analysis

In this section we provide the convergence analysis of the STORM$^+$ algorithm. We begin with the analysis in the offline case where $\sigma = 0$, and establish a convergence rate of $O(1/\sqrt{T})$ in Section 5.1 for completeness. In Section 5.2, we introduce a simplified version of STORM$^+$, with a non-adaptive momentum parameter of the form $a_{t+1} := 1/t^{2/3}$. Due to simplicity and space limitations, it is inconvenient to share the full proof of STORM$^+$, and this simplified version enables us to illustrate the main steps of the original proof. We show that this version achieves a convergence rate of $O(1/T^{1/3})$ in the stochastic case (though it does not adapt to the variance). Finally, in Section 5.1 we provide a proof sketch for STORM$^+$ in Alg. 1 that establishes the result in Theorem 1.

### 5.1 Offline Case

Here we analyze STORM$^+$ in the case where $\sigma = 0$, and demonstrate a rate of $O(1/\sqrt{T})$ for finding an approximate stationary point.

**Theorem 2.** *Let $f$ satisfy Eq.* (1), (4) *and $\bar{x}_T$ be generated after running Alg.* 1 *for $T$ iterations under deterministic oracle. Then it holds that,*

$$\mathbf{E}\|\nabla f(\bar{x}_T)\| \leq O(\sqrt{1 + L^3 + B^{9/4}}/\sqrt{T}).$$

*where we take expectation due to randomness governing the generation of $\bar{x}_T$ (see line 8 in Alg.* 1*).*

*Proof.* In the case where $\sigma = 0$ one can directly show by induction that $d_t = \bar{g}_t := \nabla f(x_t)$. So the update rule becomes $x_{t+1} = x_t - \eta_t \bar{g}_t$. Now, using the smoothness of the objective implies,

$$\Delta_{t+1} - \Delta_t = f(x_{t+1}) - f(x_t) \leq -\eta_t\|\bar{g}_t\|^2 + L\eta_t^2\|\bar{g}_t\|^2/2,$$

here we denoted $\Delta_t := f(x_t) - f(x^*)$, where $x^* \in \arg\min f(x)$. Dividing by $\eta_t$, re-arranging and summing gives,

$$\sum_{t=1}^{T} \|\bar{g}_t\|^2 \leq \frac{\Delta_1}{\eta_1} - \frac{\Delta_{T+1}}{\eta_T} + \sum_{t=2}^{T} \left( \frac{1}{\eta_t} - \frac{1}{\eta_{t-1}} \right) \Delta_t + \frac{L}{2} \sum_{t=1}^{T} \eta_t \|\bar{g}_t\|^2$$

$$\leq \frac{B}{\eta_1} + B \sum_{t=2}^{T} \left( \frac{1}{\eta_t} - \frac{1}{\eta_{t-1}} \right) + \frac{L}{2} \sum_{t=1}^{T} \frac{\|\bar{g}_t\|^2}{\left( \sum_{i=1}^{t} \|\bar{g}_i\|^2 \right)^{1/3}}$$

$$\leq \frac{B}{\eta_T} + L \left( \sum_{i=1}^{t} \|\bar{g}_i\|^2 \right)^{2/3} \leq B \left( \sum_{t=1}^{T} \|\bar{g}_t\|^2 / a_{t+1} \right)^{1/3} + L \left( \sum_{t=1}^{T} \|\bar{g}_t\|^2 \right)^{2/3}$$

$$\leq B \left( 1 + \sum_{t=1}^{T} \|\bar{g}_t\|^2 \right)^{2/9} \left( \sum_{t=1}^{T} \|\bar{g}_t\|^2 \right)^{1/3} + L \left( \sum_{t=1}^{T} \|\bar{g}_t\|^2 \right)^{2/3} \qquad (9)$$

where the second inequality uses $\eta_t = \left( \sum_{i=1}^{t} \|\bar{g}_i\|^2 / a_{i+1} \right)^{-1/3} \leq \left( \sum_{i=1}^{t} \|\bar{g}_i\|^2 \right)^{-1/3}$ which holds since $d_t = \bar{g}_t$ and $a_t \leq 1$. We also use that $\Delta_t \in [0, B]$ together with $\eta_t^{-1} - \eta_{t-1}^{-1} \geq 0$. The third inequality uses Lemma 3 below; and the last inequality uses $1/a_{t+1} \leq (1/a_{T+1}) = \left( 1 + \sum_{t=1}^{T} \|\bar{g}_t\|^2 \right)^{2/3}$, which holds since $a_t$ is monotonically non-increasing.

By treating the inequality in Eq. (9) as a polynomial of $x = \sum_{t=1}^{T} \|\bar{g}_t\|^2$, one can derive the following bound, $\sum_{t=1}^{T} \|\bar{g}_t\|^2 \leq O(1 + L^3 + B^{9/4})$. Using the definition of $\bar{x}_T$ as well as Jensen's inequality implies,

$$\mathbf{E}\|\nabla f(\bar{x}_T)\| := \mathbf{E}\|\bar{g}(\bar{x}_T)\| \leq \sqrt{\mathbf{E}\|\bar{g}(\bar{x}_T)\|^2} = \sqrt{\sum_{t=1}^{T} \|\bar{g}_t\|^2 / T} \leq O(\sqrt{1 + L^3 + B^{9/4}}/\sqrt{T}).$$

which establishes the bound. In the proof we have used the technical lemma below,

**Lemma 3.** *Let $b_1 > 0$, $b_2, ..., b_n \geq 0$ be a sequence of real numbers, $p \in (0,1)$ be a real number.*

$$\sum_{i=1}^{n} \frac{b_i}{\left( \sum_{j=1}^{i} b_j \right)^p} \leq \frac{1}{1-p} \left( \sum_{i=1}^{n} b_i \right)^{1-p}$$

$\square$

## 5.2 Stochastic Case Analysis of Simplified STORM$^+$

Here we analyze a simplified version of STORM$^+$ in the stochatic setting. While this version does not adapt to the noise variance, it exhibits the optimal rate of $O(1/T^{1/3})$ in the stochastic case, and its analysis illustrates some of the main ideas that we employ in the proof of the fully adaptive STORM$^+$ (which is more involved).

The version that we analyze here differs from STORM$^+$ in the choice of the momentum parameters. Here we choose $a_1 = 1$ and $a_{t+1} = 1/t^{2/3}$ ; $\forall t \geq 1$, in contrast to the adaptive choice that we make in Alg. 1. Note that we keep the same expression for the step size, $\eta_t = 1 / \left( \sum_{i=1}^{t} \|d_i\|^2 / a_{i+1} \right)^{1/3}$.

**Theorem 4.** *Under Assumptions in Eq. (1), (2), (3) and (4), simplified STORM$^+$ ensures,*

$$\mathbf{E}\|\nabla f(\bar{x}_T)\| = O(\sqrt{L^3 + \sigma^2 + B^{3/2}}/T^{1/3}),$$

*Proof.* The proof is composed of two parts. In the first we bound the cumulative expectation of errors $\mathbf{E} \sum_{t=1}^{T} \|\epsilon_t\|^2$, where $\epsilon_t$ is the difference between the corrected momentum $d_t$ and the exact gradient $\bar{g}_t$, i.e. $\epsilon_t = d_t - \bar{g}_t$. Thus, in the first part we relate the above sum to the sum of exact

gradients $\mathbf{E}\sum_{t=1}^{T}\|\bar{g}_t\|^2$. Then, in the second part we divide into two sub-cases the first where $\mathbf{E}\sum_{t=1}^{T}\|\epsilon_t\|^2 \le (1/2)\mathbf{E}\sum_{t=1}^{T}\|\bar{g}_t\|^2$ and its complement. In one of these sub-cases we also use the smoothness of the objective together with the update rule, similarly to what we do in Eq. (9).

**First Part: Bounding $\mathbf{E}\sum_{t=1}^{T}\|\epsilon_t\|^2$.** The update rule for $d_t$ induces the following error dynamics,

$$\epsilon_t = (1-a_t)\epsilon_{t-1} + a_t(g_t - \bar{g}_t) + (1-a_t)Z_t \tag{10}$$

where $Z_t := (g_t - \tilde{g}_{t-1}) - (\bar{g}_t - \bar{g}_{t-1})$. Letting $\mathcal{H}_t$ be the history to time $t$, i.e., $\mathcal{H}_t := \{x_1, \xi_1, \xi_2, \xi_3 \ldots, \xi_t\}$ and recalling that both $a_t$ and $x_t$ depend on history up to $t-1$, i.e., $\mathcal{H}_{t-1}$, immediately implies that $\mathbf{E}[a_t(g_t - \bar{g}_t)|\mathcal{H}_{t-1}] = \mathbf{E}[(1-a_t)Z_t|\mathcal{H}_{t-1}] = 0$, as well as $\mathbf{E}[(1-a_t)\epsilon_{t-1}|\mathcal{H}_{t-1}] = (1-a_t)\epsilon_{t-1}$.

Thus, taking the square of the above equation and then taking the expectation gives,

$$\mathbf{E}\|\epsilon_t\|^2 \le (1-a_t)^2\mathbf{E}\|\epsilon_{t-1}\|^2 + \|(1-a_t)Z_t + a_t(g_t - \bar{g}_t)\|^2$$
$$\le (1-a_t)^2\mathbf{E}\|\epsilon_{t-1}\|^2 + 2(1-a_t)^2\|Z_t\|^2 + 2a_t^2\mathbf{E}\|g_t - \bar{g}_t\|^2$$
$$\le (1-a_t)\mathbf{E}\|\epsilon_{t-1}\|^2 + 8L^2\mathbf{E}\eta_{t-1}^2\|d_{t-1}\|^2 + 2a_t^2\sigma^2 , \tag{11}$$

where the second line uses $\|b+c\|^2 \le 2\|b\|^2 + 2\|c\|^2$, and the last line uses $\mathbf{E}\|g_t - \bar{g}_t\|^2 \le \sigma^2$ and $(1-a_t) \in [0,1]$, as well as the smoothness assumption that implies $\|Z_t\| \le \|g_t - \tilde{g}_{t-1}\| + \|\bar{g}_t - \bar{g}_{t-1}\| \le 2L\|x_t - x_{t-1}\| = 2L\eta_{t-1}\|d_{t-1}\|$.

Dividing Eq. (11) by $a_t$ and re-arranging implies,

$$\mathbf{E}\|\epsilon_{t-1}\|^2 \le \frac{1}{a_t}(\mathbf{E}\|\epsilon_{t-1}\|^2 - \mathbf{E}\|\epsilon_t\|^2) + 8L^2\mathbf{E}[\eta_{t-1}^2\|d_{t-1}\|^2/a_t] + 2a_t\sigma^2 .$$

Summing the above, and using $\epsilon_0 := 0$ gives,

$$\mathbf{E}\sum_{t=1}^{T}\|\epsilon_{t-1}\|^2 \le \underbrace{-\frac{\mathbf{E}\|\epsilon_T\|^2}{a_T}}_{(A)} + \underbrace{\sum_{t=1}^{T-1}(\frac{1}{a_{t+1}} - \frac{1}{a_t})\mathbf{E}\|\epsilon_t\|^2}_{(B)} + 8L^2\underbrace{\mathbf{E}[\sum_{t=1}^{T}\eta_{t-1}^2\|d_{t-1}\|^2/a_t]}_{(C)} + 2\sigma^2\underbrace{\sum_{t=1}^{T}a_t}_{(D)} \tag{12}$$

Next we bound all the term on the RHS of the above equation:

**Bounding (A):** Since $a_T \le 1$ we can bound $-\mathbf{E}\|\epsilon_T\|^2/a_T \le -\mathbf{E}\|\epsilon_T\|^2$

**Bounding (B):** Note that $G(z) = z^{2/3}$ is a concave function in $\mathbb{R}_+$. Thus applying the gradient inequality implies that $\forall z_1, z_2 \ge 0$ we have $(z_1 + z_2)^{2/3} - z_1^{2/3} \le \frac{2}{3}z_1^{-1/3}z_2$. Hence, for all $t \ge 2$,

$$1/a_{t+1} - 1/a_t = t^{2/3} - (t-1)^{2/3} \le 2(t-1)^{-1/3}/3 \le 2/3 .$$

Moreover, $1/a_2 - 1/a_1 = 0$. These imply that $(B) \le (2/3)\mathbf{E}\sum_{t=1}^{T}\|\epsilon_t\|^2$.

**Bounding (C):** By definition of $\eta_t$ we have,

$$(C) = \mathbf{E}\sum_{t=1}^{T}\frac{\|d_{t-1}\|^2/a_t}{\left(\sum_{i=1}^{t-1}\|d_i\|^2/a_{i+1}\right)^{2/3}} \le 3\mathbf{E}\left(\sum_{t=1}^{T-1}\|d_t\|^2/a_{t+1}\right)^{1/3} \le 3T^{2/9}\left(\mathbf{E}\sum_{t=1}^{T}\|d_t\|^2\right)^{1/3} .$$

where the first inequality uses Lemma 3, and the second inequality uses $1/a_t \le 1/a_{T+1} \le T^{2/3}$ as well as Jensen's inequality with respect to the concave function $U(z) = z^{1/3}$, defined over $\mathbb{R}_+$.

**Bounding (D):** Lemma 3 immediately implies that $(D) = 1 + \sum_{t=1}^{T-1}1/t^{2/3} \le 1 + 3T^{1/3} \le 4T^{1/3}$.

Plugging these bounds into Eq. (12) and re-arranging yields,

$$\mathbf{E}\sum_{t=1}^{T}\|\epsilon_t\|^2 \le 72L^2T^{2/9}\left(\mathbf{E}\sum_{t=1}^{T}\|d_t\|^2\right)^{1/3} + 24\sigma^2T^{1/3} . \tag{13}$$

**Second Part: Bounding $\mathbf{E} \sum_{t=1}^{T} \|\bar{g}_t\|^2$.** Here we use the bound of Eq. (13) in order to bound the sum of square gradients. Let us divide into two sub-cases.

**Case 1 : Assume that $\mathbf{E} \sum_{t=1}^{T} \|\epsilon_t\|^2 \geq \frac{1}{2}\mathbf{E} \sum_{t=1}^{T} \|\bar{g}_t\|^2$.** Combining the condition of Case 1 with $\|d_t\|^2 \leq 2\|\bar{g}_t\|^2 + 2\|\epsilon_t\|^2$ (due to $d_t = \bar{g}_t + \epsilon_t$), implies that $\mathbf{E} \sum_{t=1}^{T} \|d_t\|^2 \leq 6\mathbf{E} \sum_{t=1}^{T} \|\epsilon_t\|^2$. Plugging this inside Eq. (13) yields,

$$\mathbf{E} \sum_{t=1}^{T} \|\epsilon_t\|^2 \leq 72 L^2 T^{2/9} \left( 6\mathbf{E} \sum_{t=1}^{T} \|\epsilon_t\|^2 \right)^{1/3} + 24\sigma^2 T^{1/3} \ .$$

The above immediately implies that $\mathbf{E} \sum_{t=1}^{T} \|\epsilon_t\|^2 \leq O((L^3 + \sigma^2)T^{1/3})$, and due to the condition of Case 1 we therefore have, $\mathbf{E} \sum_{t=1}^{T} \|\bar{g}_t\|^2 \leq O((L^3 + \sigma^2)T^{1/3})$. This concludes the first case.

**Case 2 : Assume that $\mathbf{E} \sum_{t=1}^{T} \|\epsilon_t\|^2 \leq \frac{1}{2}\mathbf{E} \sum_{t=1}^{T} \|\bar{g}_t\|^2$.** Combining the condition of Case 2 with $\|d_t\|^2 \leq 2\|\bar{g}_t\|^2 + 2\|\epsilon_t\|^2$ (due to $d_t = \bar{g}_t + \epsilon_t$), implies that $\mathbf{E} \sum_{t=1}^{T} \|d_t\|^2 \leq 3\mathbf{E} \sum_{t=1}^{T} \|\bar{g}_t\|^2$.

Now using the update rule $x_{t+1} = x_t - \eta_t d_t$ together with smoothness of $f(\cdot)$, one can show in a similar manner to our derivation of Eq. (9) the following bound (we defer this to the appendix),

$$\sum_{t=1}^{T} \|\bar{g}_t\|^2 \leq \sum_{t=1}^{T} \|\epsilon_t\|^2 + 2BT^{2/9} \left( \sum_{t=1}^{T} \|d_t\|^2 \right)^{1/3} + \frac{3}{2}L \left( \sum_{t=1}^{T} \|d_t\|^2 \right)^{2/3} \tag{14}$$

Taking the expectation of the above equation and plugging in $\mathbf{E} \sum_{t=1}^{T} \|d_t\|^2 \leq 3\mathbf{E} \sum_{t=1}^{T} \|\bar{g}_t\|^2$ as well as $\mathbf{E} \sum_{t=1}^{T} \|\epsilon_t\|^2 \leq \frac{1}{2}\mathbf{E} \sum_{t=1}^{T} \|\bar{g}_t\|^2$ gives,

$$\mathbf{E} \sum_{t=1}^{T} \|\bar{g}_t\|^2 \leq \frac{1}{2}\mathbf{E} \sum_{t=1}^{T} \|\bar{g}_t\|^2 + 2BT^{2/9} \left( 3\mathbf{E} \sum_{t=1}^{T} \|\bar{g}_t\|^2 \right)^{1/3} + \frac{3}{2}L \left( 3\mathbf{E} \sum_{t=1}^{T} \|\bar{g}_t\|^2 \right)^{2/3} \tag{15}$$

where we also used Jensen's inequality with respect to he concave functions $z^{1/3}$ and $z^{2/3}$ defined over $\mathbb{R}_+$. The above immediately implies, $\mathbf{E} \sum_{t=1}^{T} \|\bar{g}_t\|^2 \leq O(L^3 + B^{3/2}T^{1/3})$. This concludes the second case.

**Summary.** We have shown that $\mathbf{E} \sum_{t=1}^{T} \|\bar{g}_t\|^2 \leq O((L^3 + \sigma^2 + B^{3/2})T^{1/3})$, combining this with the definition of $\bar{x}_T$ and using Jensen's inequality similarly to what we did in the offline analysis provides,

$$\mathbf{E}\|\nabla f(\bar{x}_T)\| = O(\sqrt{L^3 + \sigma^2 + B^{3/2}}/T^{1/3}) \ ,$$

which concludes the proof. □

### 5.3 Stochastic Case Analysis of $\mathrm{STORM}^+$

Finally, we provide a sketch of the proof for the $\mathrm{STORM}^+$ algorithm appearing in Alg. 1. In a high level, the analysis follows similar lines to that of of simplified $\mathrm{STORM}^+$ 's appearing in Section 5.2.

There are two extra challenges compared to the analysis of simplified $\mathrm{STORM}^+$ :

1. Now $a_t$ is a random variable that depends on the noisy samples.
2. The differences $1/a_{t+1} - 1/a_t$ are not necessarily smaller than 1.

Recall that in the analysis appearing in Section 5.2 we used $1/a_{t+1} - 1/a_t \leq 2/3$, which was crucial to bounding term (B).

Among the tools that we use to address the first challenge is a version of Young's inequality, that we mention in the appendix. To cope with the second challenge, when we bound the expectation of $\sum_{t=1}^{T} \|\epsilon_t\|^2$, it is split into two,

$$\sum_{t=1}^{T} \|\epsilon_t\|^2 = \sum_{t=1}^{\tau^*} \|\epsilon_t\|^2 + \sum_{t=\tau^*+1}^{T} \|\epsilon_t\|^2$$

where $\tau^*$ is a time-step after which we can ensure that $1/a_{t+1} - 1/a_t \le 2/3$. Next we proceed with the proof sketch.

*Proof Sketch of Theorem 1.* The proof is composed of three parts: **(a)** In the first part we bound the cumulative expectation of errors $\mathbf{E}\sum_{t=1}^{\tau^*} \|\epsilon_t\|^2$, where $\epsilon_t := d_t - \bar{g}_t$, and $\tau^*$ is a stopping time after which we can ensure that $1/a_{t+1} - 1/a_t \le 2/3$. **(b)** In the second part we use our bound on $\mathbf{E}\sum_{t=1}^{\tau^*} \|\epsilon_t\|^2$ in order to bound the total sum of square errors, $\mathbf{E}\sum_{t=1}^{T} \|\epsilon_t\|^2$. **(c)** Then, in the last part we divide into two sub-cases the first where $\mathbf{E}\sum_{t=1}^{T} \|\epsilon_t\|^2 \le (1/2)\mathbf{E}\sum_{t=1}^{T} \|\bar{g}_t\|^2$ and its complement. In one of these sub-cases we also use the smoothness of the objective together with the update rule, similarly to what we do in Eq. (9).

**First Part: Bounding $\mathbf{E}\sum_{t=1}^{\tau^*} \|\epsilon_t\|^2$.** Recall the error dynamics of $\text{STORM}^+$ appearing in Eq. (10). Taking the square and summing up to some $\tau^* \in [T]$ enables to bound,

$$\sum_{t=1}^{\tau^*} \|\epsilon_t\|^2 \le \sum_{t=1}^{\tau^*}(1-a_t)\|\epsilon_{t-1}\|^2 + 2\sum_{t=1}^{\tau^*} \|Z_t\|^2 + 2\sum_{t=1}^{\tau^*} a_t^2 \|g_t - \bar{g}_t\|^2 + \sum_{t=1}^{\tau^*} M_t \ ,$$

where $M_t = 2\langle (1-a_t)\epsilon_{t-1}, a_t(g_t - \bar{g}_t) + (1-a_t)Z_t \rangle$ is a martingale difference sequence such that $\mathbf{E}[M_t|\mathcal{H}_{t-1}] = 0$, where $\mathcal{H}_t$ is the history to time $t$, i.e., $\mathcal{H}_t := \{x_1, \xi_1, \xi_2, \xi_3 \ldots, \xi_t\}$. Also, recall that $Z_t := (g_t - \tilde{g}_{t-1}) - (\bar{g}_t - \bar{g}_{t-1})$.

Now let us define $\beta := \min\{1, 1/G^4\}$, and $\tau^* = \max\{t \in [T] : a_t \ge \beta\}$. Recalling that $a_{t+1}$ is measurable with respect to $\mathcal{H}_t$ implies that $\tau^* \in [T]$ is a stopping time.

Re-arranging the above and using the definition of $\tau^*$ implies,

$$\beta \sum_{t=1}^{\tau^*} \|\epsilon_t\|^2 \le \|\epsilon_{\tau^*}\|^2 + \sum_{t=1}^{\tau^*-1} a_{t+1}\|\epsilon_t\|^2 \le 2\underbrace{\sum_{t=1}^{T} \|Z_t\|^2}_{\text{(i)}} + 2\underbrace{\sum_{t=1}^{T} a_t^2 \|g_t - \bar{g}_t\|^2}_{\text{(ii)}} + \underbrace{\sum_{t=1}^{\tau^*} M_t}_{\text{(iii)}}$$

where we used $\tau^* \le T$, as well as $\beta \le 1$. Next we bound the expected value of the above terms.

**Bounding (i).** As in the previous section, the smoothness property implies that $\|Z_t\|^2 \le 4L^2\eta_{t-1}^2\|d_{t-1}\|^2$. Using the expression for $\eta_{t-1}$ together with Lemma 3 enables to show,

$$\text{(i)} \le 4L^2\sum_{t=1}^{T} \frac{\|d_{t-1}\|^2}{(\sum_{i=1}^{t-1}\|d_i\|^2)^{2/3}} \le 12L^2(\sum_{t=1}^{T}\|d_t\|^2)^{1/3} \ .$$

**Bounding (ii).** One can directly show that $\mathbf{E}[a_t^2\|g_t - \bar{g}_t\|^2] \le \mathbf{E}[a_t^2\|g_t\|^2]$. Using this together with the expression for $a_t$, it is possible to show that,

$$\mathbf{E}\text{(ii)} \le \mathbf{E}\sum_{t=1}^{T} \frac{\|g_t\|^2}{(1 + \sum_{i=1}^{t-1}\|g_i\|^2)^{4/3}} \le C_1 \ .$$

where $C_1$ is a constant, and the second inequality is due to a lemma that we describe in the appendix.

**Bounding (iii).** Since $\tau^* \in [T]$ is a bounded stopping time, and $M_t$ is a martingale difference sequence, then Doob's optional stopping theorem Levin and Peres [2017] implies $\mathbf{E}\text{(iii)} = \mathbf{E}\sum_{t=1}^{\tau^*} M_t = 0$.

**Conclusion.** The above together with Jensen's inequality for $U(z) = z^{1/3}$ defined over $\mathbb{R}_+$, yields,

$$\mathbf{E}\sum_{t=1}^{\tau^*} \|\epsilon_t\|^2 \le 2C_1/\beta + 24(L^2/\beta)(\mathbf{E}\sum_{t=1}^{T}\|d_t\|^2)^{1/3} \ . \tag{16}$$

**Second Part: Bounding** $\mathbf{E}\sum_{t=1}^{T}\|\epsilon_t\|^2$**.**  Recall the error dynamics of STORM$^+$ appearing in Eq. (10). Dividing by $\sqrt{a_t}$, taking the square and summing up to some $T$ enables to bound,

$$\frac{1}{a_t}\|\epsilon_t\|^2 \leq (\frac{1}{a_t}-1)\|\epsilon_{t-1}\|^2 + 2\frac{\|Z_t\|^2}{a_t} + 2a_t\|g_t - \bar{g}_t\|^2 + Y_t$$

where $Y_t = 2\langle\frac{1-a_t}{\sqrt{a_t}}\epsilon_{t-1}, \sqrt{a_t}(g_t - \bar{g}_t) + \frac{1-a_t}{\sqrt{a_t}}Z_t\rangle$ is a martingale difference sequence such $\mathbf{E}[Y_t|\mathcal{H}_{t-1}] = 0$. Re-arranging the above and summing one can show,

$$\sum_{t=1}^{T}\|\epsilon_{t-1}\|^2 \leq \underbrace{-\frac{1}{a_T}\|\epsilon_T\|^2}_{(A)} + \underbrace{\sum_{t=1}^{T}(\frac{1}{a_{t+1}} - \frac{1}{a_t})\|\epsilon_t\|^2}_{(B)} + 2\underbrace{\sum_{t=1}^{T}\frac{\|Z_t\|^2}{a_t}}_{(C)} + 2\underbrace{\sum_{t=1}^{T}a_t\|g_t - \bar{g}_t\|^2}_{(D)} + \underbrace{\sum_{t=1}^{T}Y_t}_{(E)}$$

Now, due to the martingale property $\mathbf{E}(E) = 0$. Next, we focus on bounding term (B),
**Bounding** (B)**.** Using the definition of $\tau^*$ one can show that $1/a_{t+1} \leq 1/\tilde{\beta}$ ; $\forall t \leq \tau^*$, where $1/\tilde{\beta} := (1/\beta^{3/2} + G^2)^{2/3}$. Moreover, we can show,

$$1/a_{t+1} - 1/a_t \leq 2/3 ; \quad \forall t \geq \tau^* + 1$$

This enables to decompose and bound (B) according to $\tau^*$,

$$\sum_{t=1}^{T}(\frac{1}{a_{t+1}} - \frac{1}{a_t})\|\epsilon_t\|^2 = \sum_{t=1}^{\tau^*}(\frac{1}{a_{t+1}} - \frac{1}{a_t})\|\epsilon_t\|^2 + \sum_{t=\tau^*+1}^{T}(\frac{1}{a_{t+1}} - \frac{1}{a_t})\|\epsilon_t\|^2$$

$$\leq \frac{1}{\tilde{\beta}}\sum_{t=1}^{\tau^*}\|\epsilon_t\|^2 + \frac{2}{3}\sum_{t=\tau^*+1}^{T}\|\epsilon_t\|^2 \leq \frac{1}{\tilde{\beta}}\sum_{t=1}^{\tau^*}\|\epsilon_t\|^2 + \frac{2}{3}\sum_{t=1}^{T}\|\epsilon_t\|^2 . \quad (17)$$

This enables to use Eq. (16) to bound the expected value of term (B).

From here the analysis of the other terms and bounding $\mathbf{E}\sum_{t=1}^{T}\|\epsilon_{t-1}\|^2$ is done similarly to our analysis of simplified STORM$^+$ .

**Third Part: Bounding** $\mathbf{E}\sum_{t=1}^{T}\|\bar{g}_t\|^2$**.**  In this part we divide into two sub-cases depending whether $\mathbf{E}\sum_{t=1}^{T}\|\epsilon_t\|^2 \geq (1/2)\mathbf{E}\sum_{t=1}^{T}\|\bar{g}_t\|^2$ or not. And continue similarly to our analysis of simplified STORM$^+$ . The rest of the details appear in the appendix. $\qquad\square$

# 6   Conclusion

We have presented a novel parameter-free and adaptive algorithm for non-convex optimization that obtains the optimal rate in the setting of expectation over smooth losses while adapting to variance in gradient estimates. Our approach suggests a new way to set the learning rate and momentum jointly and adaptively throughout the learning process, which might open up new avenues to both practical and theoretical developments in this direction.

## Acknowledgments

This work has received funding from the European Research Council (ERC) under the European Union's Horizon 2020 research and innovation programme (grant agreement no 725594 - time-data); Hasler Foundation Program: Cyber Human Systems (project number 16066); the Department of the Navy, Office of Naval Research (ONR) under a grant number N62909-17-1-2111; the Swiss National Science Foundation (SNSF) under grant number 200021_178865 / 1; and the Army Research Office under Grant Number W911NF-19-1-0404. K.Y. Levy acknowledges support from the Israel Science Foundation (grant No. 447/20).

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
