where we used $\tau^* \leq T$, as well as $\beta \leq 1$. Next we bound the expected value of the above terms.

**Bounding (i).** As in the previous section, the smoothness property implies that $\|Z_t\|^2 \leq 4L^2 \eta_{t-1}^2 \|d_{t-1}\|^2$. Using the expression for $\eta_{t-1}$ together with Lemma 3 enables to show,

$$(i) \leq 4L^2 \sum_{t=1}^{T} \frac{\|d_{t-1}\|^2}{(\sum_{i=1}^{t-1} \|d_i\|^2)^{2/3}} \leq 12L^2 (\sum_{t=1}^{T} \|d_t\|^2)^{1/3} .$$

**Bounding (ii).** One can directly show that $\mathbf{E}[a_t^2 \|g_t - \bar{g}_t\|^2] \leq \mathbf{E}[a_t^2 \|g_t\|^2]$. Using this together with the expression for $a_t$, it is possible to show that,

$$\mathbf{E}(ii) \leq \mathbf{E} \sum_{t=1}^{T} \frac{\|g_t\|^2}{(1 + \sum_{i=1}^{t-1} \|g_i\|^2)^{4/3}} \leq C_1 .$$

where $C_1$ is a constant, and the second inequality is due to a lemma that we describe in the appendix.

**Bounding (iii).** Since $\tau^* \in [T]$ is a bounded stopping time, and $M_t$ is a martingale difference sequence, then Doob's optional stopping theorem Levin and Peres [2017] implies $\mathbf{E}(iii) = \mathbf{E} \sum_{t=1}^{\tau^*} M_t = 0$.

**Conclusion.** The above together with Jensen's inequality for $U(z) = z^{1/3}$ defined over $\mathbb{R}_+$, yields,

$$\mathbf{E} \sum_{t=1}^{\tau^*} \|\epsilon_t\|^2 \leq 2C_1/\beta + 24(L^2/\beta)(\mathbf{E} \sum_{t=1}^{T} \|d_t\|^2)^{1/3} . \tag{16}$$

**Second Part: Bounding $\mathbf{E}\sum_{t=1}^{T}\|\epsilon_t\|^2$.** Recall the error dynamics of STORM$^+$ appearing in Eq. (10). Dividing by $\sqrt{a_t}$, taking the square and summing up to some $T$ enables to bound,

$$\frac{1}{a_t}\|\epsilon_t\|^2 \leq (\frac{1}{a_t}-1)\|\epsilon_{t-1}\|^2 + 2\frac{\|Z_t\|^2}{a_t} + 2a_t\|g_t - \bar{g}_t\|^2 + Y_t$$

where $Y_t = 2\langle \frac{1-a_t}{\sqrt{a_t}}\epsilon_{t-1}, \sqrt{a_t}(g_t - \bar{g}_t) + \frac{1-a_t}{\sqrt{a_t}}Z_t\rangle$ is a martingale difference sequence such $\mathbf{E}[Y_t|\mathcal{H}_{t-1}] = 0$. Re-arranging the above and summing one can show,

$$\sum_{t=1}^{T}\|\epsilon_{t-1}\|^2 \leq \underbrace{-\frac{1}{a_T}\|\epsilon_T\|^2}_{(A)} + \underbrace{\sum_{t=1}^{T}(\frac{1}{a_{t+1}} - \frac{1}{a_t})\|\epsilon_t\|^2}_{(B)} + 2\underbrace{\sum_{t=1}^{T}\frac{\|Z_t\|^2}{a_t}}_{(C)} + 2\underbrace{\sum_{t=1}^{T}a_t\|g_t - \bar{g}_t\|^2}_{(D)} + \underbrace{\sum_{t=1}^{T}Y_t}_{(E)}$$

Now, due to the martingale property $\mathbf{E}(\mathrm{E}) = 0$. Next, we focus on bounding term (B),

**Bounding (B).** Using the definition of $\tau^*$ one can show that $1/a_{t+1} \leq 1/\tilde{\beta}$ ; $\forall t \leq \tau^*$, where $1/\tilde{\beta} := (1/\beta^{3/2} + G^2)^{2/3}$. Moreover, we can show,

$$1/a_{t+1} - 1/a_t \leq 2/3 ; \quad \forall t \geq \tau^* + 1$$

This enables to decompose and bound (B) according to $\tau^*$,

$$\sum_{t=1}^{T}(\frac{1}{a_{t+1}} - \frac{1}{a_t})\|\epsilon_t\|^2 = \sum_{t=1}^{\tau^*}(\frac{1}{a_{t+1}} - \frac{1}{a_t})\|\epsilon_t\|^2 + \sum_{t=\tau^*+1}^{T}(\frac{1}{a_{t+1}} - \frac{1}{a_t})\|\epsilon_t\|^2$$

$$\leq \frac{1}{\tilde{\beta}}\sum_{t=1}^{\tau^*}\|\epsilon_t\|^2 + \frac{2}{3}\sum_{t=\tau^*+1}^{T}\|\epsilon_t\|^2 \leq \frac{1}{\tilde{\beta}}\sum_{t=1}^{\tau^*}\|\epsilon_t\|^2 + \frac{2}{3}\sum_{t=1}^{T}\|\epsilon_t\|^2 . \quad (17)$$

This enables to use Eq. (16) to bound the expected value of term (B).

From here the analysis of the other terms and bounding $\mathbf{E}\sum_{t=1}^{T}\|\epsilon_{t-1}\|^2$ is done similarly to our analysis of simplified STORM$^+$ .

**Third Part: Bounding $\mathbf{E}\sum_{t=1}^{T}\|\bar{g}_t\|^2$.** In this part we divide into two sub-cases depending whether $\mathbf{E}\sum_{t=1}^{T}\|\epsilon_t\|^2 \geq (1/2)\mathbf{E}\sum_{t=1}^{T}\|\bar{g}_t\|^2$ or not. And continue similarly to our analysis of simplified STORM$^+$ . The rest of the details appear in the appendix. $\qquad\square$

# 6  Conclusion

We have presented a novel parameter-free and adaptive algorithm for non-convex optimization that obtains the optimal rate in the setting of expectation over smooth losses while adapting to variance in gradient estimates. Our approach suggests a new way to set the learning rate and momentum jointly and adaptively throughout the learning process, which might open up new avenues to both practical and theoretical developments in this direction.

# Acknowledgments

This work has received funding from the European Research Council (ERC) under the European Union's Horizon 2020 research and innovation programme (grant agreement no 725594 - time-data); Hasler Foundation Program: Cyber Human Systems (project number 16066); the Department of the Navy, Office of Naval Research (ONR) under a grant number N62909-17-1-2111; the Swiss National Science Foundation (SNSF) under grant number 200021_178865 / 1; and the Army Research Office under Grant Number W911NF-19-1-0404. K.Y. Levy acknowledges support from the Israel Science Foundation (grant No. 447/20).

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

# A  Appendix

## A.1  Proofs for Section 5.2

### A.1.1  Proof of Equation (14)

We will use the following lemma that we prove in Section A.1.3,

**Lemma 5.** *For both* $\mathrm{STORM}^+$ *and simplified* $\mathrm{STORM}^+$ *the following holds,*

$$\sum_{t=1}^{T} \|\bar{g}_t\|^2 \leq \sum_{t=1}^{T} \|\epsilon_t\|^2 + 2B a_{T+1}^{-1/3} (\sum_{t=1}^{T} \|d_t\|^2)^{1/3} + \frac{3}{2} L (\sum_{t=1}^{T} \|d_t\|^2)^{2/3}$$

*Eq. (14) directly follows from this lemma by taking* $a_{T+1} = 1/T^{2/3}$.

### A.1.2  Proof of Lemma 3

We will prove the lemma by induction on $n$. The proof relies on the arguments in [McMahan and Streeter, 2010] and generalizes it for any $p \in (0, 1)$.

*Proof.* For the base case of $n = 1$, we can easily show that the hypothesis holds.

$$\frac{b_1}{b_1^p} = b_1^{1-p} \leq \frac{1}{1-p} b_1^{1-p}$$

Now, assuming that the hypothesis holds for some arbitrary number $n - 1 > 1$, we want to show that it holds for $n$, too. Let us define $Z = \sum_{t=1}^{n} b_t$ and $x = b_n$. Then, using the inductive hypothesis for $n - 1$,

$$\sum_{t=1}^{n} \frac{b_n}{\left(\sum_{i=1}^{t} b_i\right)^p} \leq \frac{1}{1-p} \left(\sum_{t=1}^{n-1} b_t\right)^{1-p} + \frac{b_n}{\left(\sum_{t=1}^{n} b_t\right)^p}$$

$$= \frac{1}{1-p} (Z - x)^{1-p} + \frac{x}{Z^p}$$

Let us denote $h(x) = \frac{1}{1-p}(Z - x)^{1-p} + \frac{x}{Z^p}$ is concave in $x$. What we need to show is that, for any choice of allowable $x$, $h(x) \leq \frac{1}{1-p} Z^{1-p}$. Specifically, we want to prove that

$$\max_{0 \leq x < Z} h(x) \leq \frac{1}{1-p} Z^{1-p}$$

First, observe that $h(x)$ is a concave function, hence at the maximum the derivative evaluates to zero. Our aim is to find such $x$. Taking derivative wrt $x$,

$$\frac{dh(x)}{dx} = \frac{1}{Z^p} - \frac{1}{(Z-x)^p},$$

which evaluates to zero when $x = 0$. Hence,

$$\max_{0 \leq x < Z} h(x) = h(0) = \frac{1}{1-p} Z^{1-p} = \frac{1}{1-p} \left(\sum_{t=1}^{n} b_t\right)^{1-p}$$

which implies that the hypothesis is true:

$$\sum_{t=1}^{n} \frac{b_t}{\left(\sum_{i=1}^{t} b_i\right)^p} \leq \frac{1}{1-p} \left(\sum_{t=1}^{n} b_t\right)^{1-p}.$$

$\square$

### A.1.3 Proof of Lemma 5

Using smoothness together with the update rule implies,

$$\Delta_{t+1} - \Delta_t = f(x_{t+1}) - f(x_t) \le -\eta_t \bar{g}_t^\top d_t + \frac{L\eta_t^2}{2} \|d_t\|^2$$

$$= -\eta_t \|\bar{g}_t\|^2 - \eta_t \bar{g}_t^\top \epsilon_t + \frac{L\eta_t^2}{2} \|d_t\|^2$$

$$\le -\eta_t \|\bar{g}_t\|^2 + \frac{\eta_t}{2} \|\bar{g}_t\|^2 + \frac{\eta_t}{2} \|\epsilon_t\|^2 + \frac{L\eta_t^2}{2} \|d_t\|^2 \,,$$

where we defined $\Delta_t := f(x_t) - f(x^*)$. The second line above uses $d_t = \bar{g}_t + \epsilon_t$, and the third line uses $z^\top y \le \frac{1}{2}(\|z\|^2 + \|y\|^2)$.

Re-arranging the above we get,

$$\|\bar{g}_t\|^2 \le \|\epsilon_t\|^2 + \frac{2}{\eta_t}(\Delta_t - \Delta_{t+1}) + L\eta_t \|d_t\|^2$$

Summing over $t$ gives,

$$\sum_{t=1}^T \|\bar{g}_t\|^2 \le \sum_{t=1}^T \|\epsilon_t\|^2 - \frac{2}{\eta_T}\Delta_{T+1} + 2\sum_{t=1}^T (\frac{1}{\eta_t} - \frac{1}{\eta_{t-1}})\Delta_t + L\sum_{t=1}^T \eta_t \|d_t\|^2$$

$$\le \sum_{t=1}^T \|\epsilon_t\|^2 + 2B\sum_{t=1}^T (\frac{1}{\eta_t} - \frac{1}{\eta_{t-1}}) + L\sum_{t=1}^T \frac{\|d_t\|^2}{(\sum_{i=1}^t \|d_t\|^2)^{1/3}}$$

$$\le \sum_{t=1}^T \|\epsilon_t\|^2 + 2B\frac{1}{\eta_T} + \frac{3}{2}L(\sum_{t=1}^T \|d_t\|^2)^{2/3}$$

$$\le \sum_{t=1}^T \|\epsilon_t\|^2 + 2B(\sum_{t=1}^T \|d_t\|^2/a_{t+1})^{1/3} + \frac{3}{2}L(\sum_{t=1}^T \|d_t\|^2)^{2/3}$$

$$\le \sum_{t=1}^T \|\epsilon_t\|^2 + 2B(1/a_{T+1})^{1/3}(\sum_{t=1}^T \|d_t\|^2)^{1/3} + \frac{3}{2}L(\sum_{t=1}^T \|d_t\|^2)^{2/3} \qquad (18)$$

The second line uses $\Delta_t \in [0, B]$, the third line uses Lemma 3, and the last line uses the fact that $a_t$ is monotonically decreasing.

## A.2 Full Analysis of $\text{STORM}^+$ (Algorithm 1)

The proof is composed of three parts: **(a)** In the first part we bound the cumulative expectation of errors $\mathbf{E}\sum_{t=1}^{\tau^*}\|\epsilon_t\|^2$, where $\epsilon_t := d_t - \bar{g}_t$, and $\tau^*$ is a stopping time after which we can ensure that $1/a_{t+1} - 1/a_t \leq 2/3$. **(b)** In the second part we use our bound on $\mathbf{E}\sum_{t=1}^{\tau^*}\|\epsilon_t\|^2$ in order to bound the total sum of square errors, $\mathbf{E}\sum_{t=1}^{T}\|\epsilon_t\|^2$. **(c)** Then, in the last part we divide into two sub-cases the first where $\mathbf{E}\sum_{t=1}^{T}\|\epsilon_t\|^2 \leq (1/2)\mathbf{E}\sum_{t=1}^{T}\|\bar{g}_t\|^2$ and its complement. In one of these sub-cases we also use the smoothness of the objective together with the update rule, similarly to what we do in Eq. (9).

The different parts of the proof are divided between Section A.2.1, A.2.2 , and A.2.3.

### A.2.1 First Part: Bounding $\mathbf{E}\sum_{t=1}^{\tau^*}\|\epsilon_t\|^2$.

The update rule for $d_t$ induces the following error dynamics,

$$\epsilon_t = (1-a_t)\epsilon_{t-1} + a_t(g_t - \bar{g}_t) + (1-a_t)Z_t \tag{19}$$

where $Z_t := (g_t - \tilde{g}_{t-1}) - (\bar{g}_t - \bar{g}_{t-1})$.

Taking the square and summing up to some $\tau^* \in [T]$ enables to bound,

$$\sum_{t=1}^{\tau^*}\|\epsilon_t\|^2 \leq \sum_{t=1}^{\tau^*}(1-a_t)^2\|\epsilon_{t-1}\|^2 + \sum_{t=1}^{\tau^*}\|(1-a_t)Z_t + a_t(g_t-\bar{g}_t)\|^2 + \sum_{t=1}^{\tau^*}M_t$$

$$\leq \sum_{t=1}^{\tau^*}(1-a_t)\|\epsilon_{t-1}\|^2 + 2\sum_{t=1}^{\tau^*}\|Z_t\|^2 + 2\sum_{t=1}^{\tau^*}a_t^2\|g_t-\bar{g}_t\|^2 + \sum_{t=1}^{\tau^*}M_t \,,$$

where we used $\|b+c\|^2 \leq 2\|b\|^2 + 2\|b\|^2$, as well as $(1-a_t) \leq 1$. We have defined $\{M_t := 2(1-a_t)\epsilon_{t-1}^\top((1-a_t)Z_t + a_t(g_t - \bar{g}_t))\}_{t\in[T]}$, and it is immediate to verify that $\{M_t\}_{t\in[T]}$ is a martingale difference sequence such $\mathbf{E}[M_t|\mathcal{H}_{t-1}] = 0$, where $\mathcal{H}_t$ is the history to time $t$, i.e., $\mathcal{H}_t := \{x_1, \xi_1, \xi_2, \xi_3 \dots, \xi_t\}$.

Now let us define $\beta := \min\{1, 1/G^4\}$, and $\tau^* = \max\{t \in [T] : a_t \geq \beta\}$. Recalling that $a_{t+1}$ is measurable with respect to $\mathcal{H}_t$ implies that $\tau^* \in [T]$ is a stopping time.

Re-arranging the above and using the definition of $\tau^*$ implies,

$$\beta\sum_{t=1}^{\tau^*}\|\epsilon_t\|^2 \leq \|\epsilon_{\tau^*}\|^2 + \sum_{t=1}^{\tau^*-1}a_{t+1}\|\epsilon_t\|^2 \leq 2\underbrace{\sum_{t=1}^{T}\|Z_t\|^2}_{(i)} + 2\underbrace{\sum_{t=1}^{T}a_t^2\|g_t-\bar{g}_t\|^2}_{(ii)} + \underbrace{\sum_{t=1}^{\tau^*}M_t}_{(iii)} \tag{20}$$

where we used $\tau^* \leq T$, as well as $\beta \leq 1$. Next we bound the expected value of the above terms.

**Bounding** (i). Using smoothness property implies that $\|Z_t\| \leq 2L\|x_t - x_{t-1}\| = 2L\eta_{t-1}\|d_{t-1}\|$. Using the expression for $\eta_{t-1}$ together with Lemma 3 enables to show,

$$(i) \leq 4L^2\sum_{t=1}^{T}\frac{\|d_{t-1}\|^2}{(\sum_{i=1}^{t-1}\|d_i\|^2)^{2/3}} \leq 12L^2(\sum_{t=1}^{T}\|d_t\|^2)^{1/3} \,.$$

where the first inequality uses $\eta_t = 1/\left(\sum_{i=1}^{t}\|d_i\|^2/a_{i+1}\right)^{1/3} \leq 1/\left(\sum_{i=1}^{t}\|d_i\|^2\right)^{1/3}$.

**Bounding** (ii). Since $\mathbf{E}[g_t|H_{t-1}] = \bar{g}_t$ and $a_t$ is measurable with respect to $\mathcal{H}_{t-1}$ it follows that

$$\mathbf{E}[a_t^2\|g_t-\bar{g}_t\|^2] \leq \mathbf{E}[a_t^2(\|g_t\|^2 - \|\bar{g}_t\|^2)] \leq \mathbf{E}[a_t^2\|g_t\|^2]$$

Using this together with the expression for $a_t$, it is possible to show that,

$$\mathbf{E}(ii) \leq \mathbf{E}\sum_{t=1}^{T}\frac{\|g_t\|^2}{(1+\sum_{i=1}^{t-1}\|g_i\|^2)^{4/3}} \leq C_1 \,.$$

where $C_1 := 12 + 2G^2$, and the last inequality is due to the following lemma (recall $G$ is a bound on the gradient norms),

**Lemma 6.** *For any non-negative real numbers $a_1, \ldots, a_n \in [0, a_{\max}]$,*

$$\sum_{i=1}^{n} \frac{a_i}{(1 + \sum_{j=1}^{i-1} a_j)^{4/3}} \leq 12 + 2a_{\max} .$$

We prove this lemma in Appendix A.3.1.

**Bounding** (iii). Since $\tau^* \in [T]$ is a bounded stopping time, and $M_t$ is a martingale difference sequence, then Doob's optional stopping theorem Levin and Peres [2017] implies $\mathbf{E}(\text{iii}) = \mathbf{E} \sum_{t=1}^{\tau^*} M_t = 0$.

**Conclusion.** Combining the above bounds inside Eq. (20) together with Jensen's inequality for $U(z) = z^{1/3}$ defined over $\mathbb{R}_+$, yields,

$$\mathbf{E} \sum_{t=1}^{\tau^*} \|\epsilon_t\|^2 \leq 2C_1/\beta + 24(L^2/\beta)(\mathbf{E} \sum_{t=1}^{T} \|d_t\|^2)^{1/3} . \tag{21}$$

### A.2.2 Second Part: Bounding $\mathbf{E} \sum_{t=1}^{T} \|\epsilon_t\|^2$.

Recall the error dynamics of $\text{STORM}^+$ appearing in Eq. (19). Dividing by $\sqrt{a_t}$, and taking the square gives,

$$\frac{1}{a_t}\|\epsilon_t\|^2 = (\frac{1}{a_t} - 2 + a_t)\|\epsilon_{t-1}\|^2 + \|(1 - a_t)\frac{Z_t}{\sqrt{a_t}} + \sqrt{a_t}(g_t - \bar{g}_t)\|^2 + Y_t$$

$$\leq (\frac{1}{a_t} - 1)\|\epsilon_{t-1}\|^2 + 2\frac{\|Z_t\|^2}{a_t} + 2a_t\|g_t - \bar{g}_t\|^2 + Y_t ,$$

where we used $a_t \in [0, 1]$, and $(1 - a_t) \in [0, 1]$, as well as $\|b + c\|^2 \leq 2\|b\|^2 + 2\|c\|^2$. We also defined $Y_t := 2(\frac{1}{\sqrt{a_t}} - \sqrt{a_t})\epsilon_{t-1}^\top \left((1 - a_t)\frac{Z_t}{\sqrt{a_t}} + \sqrt{a_t}(g_t - \bar{g}_t)\right)$. Note that $\mathbf{E}[Y_t|\mathcal{H}_{t-1}] = 0$; therefore $Y_t$ is a martingale difference sequence.

Re-arranging the above and summing gives,

$$\sum_{t=1}^{T} \|\epsilon_{t-1}\|^2 \leq \underbrace{-\frac{1}{a_T}\|\epsilon_T\|^2}_{(A)} + \underbrace{\sum_{t=1}^{T}(\frac{1}{a_{t+1}} - \frac{1}{a_t})\|\epsilon_t\|^2}_{(B)} + 2\underbrace{\sum_{t=1}^{T}\frac{\|Z_t\|^2}{a_t}}_{(C)} + 2\underbrace{\sum_{t=1}^{T} a_t\|g_t - \bar{g}_t\|^2}_{(D)} + \underbrace{\sum_{t=1}^{T} Y_t}_{(E)} . \tag{22}$$

Next, we bound the expected value if each of the above terms.

**Bounding** (A): Since $a_T \leq 1$ we can bound $-\mathbf{E} \|\epsilon_T\|^2/a_T \leq -\mathbf{E} \|\epsilon_T\|^2$

**Bounding** (B). We will use the following lemma which we prove in Section A.3.3,

**Lemma 7.** *The following holds,*

$$1/a_{t+1} \leq 1/\tilde{\beta} \; ; \forall t \leq \tau^*$$

*where $1/\tilde{\beta} := (1/\beta^{3/2} + G^2)^{2/3} .$*

*Moreover,*

$$1/a_{t+1} - 1/a_t \leq 2/3 ; \quad \forall t \geq \tau^* + 1$$

Lemma 7 enables to decompose and bound (B) as follows,

$$\sum_{t=1}^{T}(\frac{1}{a_{t+1}} - \frac{1}{a_t})\|\epsilon_t\|^2 = \sum_{t=1}^{\tau^*}(\frac{1}{a_{t+1}} - \frac{1}{a_t})\|\epsilon_t\|^2 + \sum_{t=\tau^*+1}^{T}(\frac{1}{a_{t+1}} - \frac{1}{a_t})\|\epsilon_t\|^2$$

$$\leq \frac{1}{\tilde{\beta}}\sum_{t=1}^{\tau^*}\|\epsilon_t\|^2 + \frac{2}{3}\sum_{t=\tau^*+1}^{T}\|\epsilon_t\|^2 \leq \frac{1}{\tilde{\beta}}\sum_{t=1}^{\tau^*}\|\epsilon_t\|^2 + \frac{2}{3}\sum_{t=1}^{T}\|\epsilon_t\|^2 . \tag{23}$$

Thus,

$$\mathbf{E}(\mathrm{B}) \le \mathbf{E} \frac{1}{\tilde{\beta}} \sum_{t=1}^{\tau^*} \|\epsilon_t\|^2 + \frac{2}{3} \mathbf{E} \sum_{t=1}^{T} \|\epsilon_t\|^2$$

$$\le \frac{2C_1}{\beta\tilde{\beta}} + 24 \frac{L^2}{\beta\tilde{\beta}} (\mathbf{E} \sum_{t=1}^{T} \|d_t\|^2)^{1/3} + \frac{2}{3} \mathbf{E} \sum_{t=1}^{T} \|\epsilon_t\|^2 \tag{24}$$

where we have used Eq. (21).

**Bounding** (C). Recalling that $\|Z_t\| \le 2L\|x_t - x_{t-1}\| = 2L\eta_{t-1}\|d_{t-1}\|$, and using the expression for $\eta_{t-1}$ together with Lemma 3 enables to show,

Thus,

$$\sum_{t=1}^{T} \frac{\|Z_t\|^2}{a_t} \le 4L^2 \sum_{t=1}^{T} \eta_{t-1}^2 \|d_{t-1}\|^2 / a_t$$

$$= 4L^2 \sum_{t=1}^{T} \frac{\|d_{t-1}\|^2 / a_t}{\left(\sum_{i=1}^{t-1} \|d_i\|^2 / a_{i+1}\right)^{2/3}}$$

$$\le 12L^2 \left(\sum_{t=1}^{T-1} \|d_t\|^2 / a_{t+1}\right)^{1/3}$$

$$\le 12L^2 \frac{1}{a_T^{1/3}} \left(\sum_{t=1}^{T-1} \|d_t\|^2\right)^{1/3}$$

$$\le 12L^2 \left(1 + \sum_{t=1}^{T} \|g_t\|^2\right)^{2/9} \left(\sum_{t=1}^{T} \|d_t\|^2\right)^{1/3}, \tag{25}$$

where we used the fact that $a_t$ is non-increasing.

Now, let us recall Young's inequality which states that for any $a, b > 0$, and $p, q > 1: \frac{1}{p} + \frac{1}{q} = 1$ we have $ab \le a^p/p + b^q/q$. This implies that for any $a, b, \rho > 0$ and $p = \frac{3}{2}, q = 3$, we have,

$$a^{2/9} b^{1/3} = (a\rho^{9/2})^{2/9} (b/\rho^3)^{1/3} \le \frac{(a\rho^{9/2})^{2p/9}}{p} + \frac{(b/\rho^3)^{q/3}}{q} = \frac{2}{3} a^{1/3} \rho^{3/2} + \frac{b}{3\rho^3} \tag{26}$$

Thus, taking $\rho = (512L^2)^{1/3}$, $a = 1 + \sum_{t=1}^{T} \|g_t\|^2$, $b = \sum_{t=1}^{T} \|d_t\|^2$, and using Young's inequality inside Eq. (25) implies,

$$\sum_{t=1}^{T} \frac{\|Z_t\|^2}{a_t} \le 512L^3 \left(1 + \sum_{t=1}^{T} \|g_t\|^2\right)^{1/3} + \frac{1}{128} \sum_{t=1}^{T} \|d_t\|^2 \tag{27}$$

**Bounding Term** (D): Note that $a_t$ is measurable with respect to $\mathcal{H}_{t-1}$, and $\mathbf{E}[g_t|\mathcal{H}_{t-1}] = \bar{g}_t$, therefore using smoothing gives,

$$\mathbf{E}[a_t\|g_t - \bar{g}_t\|^2] = \mathbf{E}[a_t(\|g_t\|^2 - \|\bar{g}_t\|^2)] \le \mathbf{E}[a_t\|g_t\|^2]$$

Thus,

$$
\mathbf{E}[(\text{D})] := \mathbf{E}\sum_{t=1}^{T} a_t \|g_t - \bar{g}_t\|^2
$$

$$
\leq \mathbf{E}\sum_{t=1}^{T} a_t \|g_t\|^2
$$

$$
= \mathbf{E}\sum_{t=1}^{T} \frac{\|g_t\|^2}{(1 + \sum_{i=1}^{t-1}\|g_i\|^2)^{2/3}}
$$

$$
\leq G^2 + 6\mathbf{E}\left(1 + \sum_{t=1}^{T}\|g_t\|^2\right)^{1/3},
$$

where the last line is due to the following lemma which is a modified and time-shifted version of Lemma 3. We defer its proof to Appendix A.3.4.

**Lemma 8.** *Let $b_1, ..., b_n \in (0, b]$ be a sequence of non-negative real numbers for some positive real number $b$, $b_0 > 0$ and $p \in (0, 1)$ a rational number. Then,*

$$
\sum_{i=1}^{n} \frac{b_i}{\left(b_0 + \sum_{j=1}^{i-1} b_j\right)^p} \leq \frac{b}{(b_0)^p} + \frac{2}{1-p}\left(b_0 + \sum_{i=1}^{n} b_i\right)^{1-p}
$$

**Bounding Term** (E)**:**   Since $\{Y_t\}_{t\in[T]}$ is a martingale difference sequence we have,

$$
\mathbf{E}(\text{E}) = \mathbf{E}\sum_{t=1}^{T} Y_t = 0 .
$$

**To Summarize:**   Combining the above bounds inside Eq. (22) we conclude that,

$$
\frac{1}{3}\mathbf{E}\sum_{t=1}^{T}\|\epsilon_t\|^2 \leq \frac{24L^2}{\beta\tilde{\beta}}\mathbf{E}(\sum_{t=1}^{T}\|d_t\|^2)^{1/3} + \frac{2C_1}{\beta\tilde{\beta}} + 2G^2
$$

$$
+ (1024L^3 + 12)\mathbf{E}\left(1 + \sum_{t=1}^{T}\|g_t\|^2\right)^{1/3} + \frac{1}{64}\mathbf{E}\sum_{t=1}^{T}\|d_t\|^2
$$

$$
\leq \frac{24L^2}{\beta\tilde{\beta}}(\mathbf{E}\sum_{t=1}^{T}\|d_t\|^2)^{1/3} + \frac{2C_1}{\beta\tilde{\beta}} + 2G^2
$$

$$
+ (1024L^3 + 12)\left(1 + \mathbf{E}\sum_{t=1}^{T}\|g_t\|^2\right)^{1/3} + \frac{1}{64}\mathbf{E}\sum_{t=1}^{T}\|d_t\|^2 \qquad (28)
$$

where we have used Jensen's inequality for the concave function $G(z) = z^{1/3}\,; z \geq 0$.

### A.2.3   Final Part of the Proof

We divide the final part of the proof into two subcases:

**Case 1: Assume $\mathbf{E}\sum_{t=1}^{T}\|\epsilon_t\|^2 \geq (1/2)\mathbf{E}\sum_{t=1}^{T}\|\bar{g}_t\|^2$.** Using the condition of this subcase implies

$$
\mathbf{E}\sum_{t}\|d_t\|^2 \leq 2\mathbf{E}\sum_{t=1}^{T}\|\bar{g}_t\|^2 + 2\mathbf{E}\sum_{t=1}^{T}\|\epsilon_t\|^2 \leq 6\mathbf{E}\sum_{t=1}^{T}\|\epsilon_t\|^2
$$

Plugging this into Eq. (28) gives,

$$
\frac{1}{3}\mathbf{E}\sum_{t=1}^{T}\|\epsilon_t\|^2 \leq \frac{24L^2}{\beta\tilde{\beta}}(6\mathbf{E}\sum_{t=1}^{T}\|\epsilon_t\|^2)^{1/3} + \frac{2C_1}{\beta\tilde{\beta}} + 2G^2
$$

$$
+ (1024L^3 + 12)\left(1 + \sigma^2 T + \mathbf{E}\sum_{t=1}^{T}\|\bar{g}_t\|^2\right)^{1/3} + \frac{6}{64}\mathbf{E}\sum_{t=1}^{T}\|\epsilon_t\|^2
$$

where the first line uses $\mathbf{E}\|g_t\|^2 = \mathbf{E}\|\bar{g}_t\|^2 + \mathbf{E}\|g_t - \bar{g}_t\|^2 \le \mathbf{E}\|\bar{g}_t\|^2 + \sigma^2$.

Re-arranging and using $\mathbf{E}\sum_{t=1}^{T}\|\bar{g}_t\|^2 \le 2\mathbf{E}\sum_{t=1}^{T}\|\epsilon_t\|^2$ gives,

$$\frac{1}{5}\mathbf{E}\sum_{t=1}^{T}\|\epsilon_t\|^2 \le \frac{24L^2}{\beta\tilde{\beta}}(6\mathbf{E}\sum_{t=1}^{T}\|\epsilon_t\|^2)^{1/3} + \frac{2C_1}{\beta\tilde{\beta}} + 2G^2$$

$$+ (1024L^3 + 12)\left(1 + \sigma^2 T + 2\mathbf{E}\sum_{t=1}^{T}\|\epsilon_t\|^2\right)^{1/3}$$

And the above implies,

$$\mathbf{E}\sum_{t=1}^{T}\|\bar{g}_t\|^2 \le 2\mathbf{E}\sum_{t=1}^{T}\|\epsilon_t\|^2 \le O\left(1 + \frac{C_1}{\beta\tilde{\beta}} + \left(\frac{L^2}{\beta\tilde{\beta}}\right)^{3/2} + G^2 + L^3 + L^{9/2} + L^3\sigma^{2/3}T^{1/3}\right) \tag{29}$$

**Case 2: Assume $\mathbf{E}\sum_{t=1}^{T}\|\epsilon_t\|^2 \le (1/2)\mathbf{E}\sum_{t=1}^{T}\|\bar{g}_t\|^2$.** Using Lemma 5 we get,

$$\sum_{t=1}^{T}\|\bar{g}_t\|^2 \le \sum_{t=1}^{T}\|\epsilon_t\|^2 + 2B(1 + \sum_{t=1}^{T}\|g_t\|^2)^{2/9}(\sum_{t=1}^{T}\|d_t\|^2)^{1/3} + \frac{3}{2}L(\sum_{t=1}^{T}\|d_t\|^2)^{2/3}$$

$$\le \sum_{t=1}^{T}\|\epsilon_t\|^2 + \frac{3}{2}L(\sum_{t=1}^{T}\|d_t\|^2)^{2/3} + 20B^{3/2}(1 + \sum_{t=1}^{T}\|g_t\|^2)^{1/3} + \frac{1}{64}\sum_{t=1}^{T}\|d_t\|^2 \tag{30}$$

where the second line uses a version of Young's inequality appearing in Eq. (26) together with taking $\rho := (128B/3)^{1/3}$, $a := 1 + \sum_{t=1}^{T}\|g_t\|^2$, and $b := \sum_{t=1}^{T}\|d_t\|^2$.

Using the condition of this subcase implies

$$\mathbf{E}\sum_{t}\|d_t\|^2 \le 2\mathbf{E}\sum_{t=1}^{T}\|\bar{g}_t\|^2 + 2\mathbf{E}\sum_{t=1}^{T}\|\epsilon_t\|^2 \le 3\mathbf{E}\sum_{t=1}^{T}\|\bar{g}_t\|^2$$

Taking expectation of Eq. (30), and using the above together with the condition gives,

$$\mathbf{E}\sum_{t=1}^{T}\|\bar{g}_t\|^2 \le \mathbf{E}\sum_{t=1}^{T}\|\epsilon_t\|^2 + \frac{3}{2}L(\mathbf{E}\sum_{t=1}^{T}\|d_t\|^2)^{2/3} + 20B^{3/2}(1 + \mathbf{E}\sum_{t=1}^{T}\|g_t\|^2)^{1/3} + \frac{1}{64}\mathbf{E}\sum_{t=1}^{T}\|d_t\|^2$$

$$\le \left(\frac{1}{2} + \frac{3}{64}\right)\mathbf{E}\sum_{t=1}^{T}\|\bar{g}_t\|^2 + \frac{3}{2}L(3\mathbf{E}\sum_{t=1}^{T}\|\bar{g}_t\|^2)^{2/3} + 20B^{3/2}(1 + \sigma^2 T + \mathbf{E}\sum_{t=1}^{T}\|\bar{g}_t\|^2)^{1/3}$$

where we have used Jensen's inequality for the functions $z^{1/3}, z^{2/3}$ defined over $\mathbb{R}_+$, We also uses $\mathbf{E}\|g_t\|^2 = \mathbf{E}\|\bar{g}_t\|^2 + \mathbf{E}\|g_t - \bar{g}_t\|^2 \le \mathbf{E}\|\bar{g}_t\|^2 + \sigma^2$.

Re-arranging the above we conclude that,

$$\mathbf{E}\sum_{t=1}^{T}\|\bar{g}_t\|^2 \le 6L(3\mathbf{E}\sum_{t=1}^{T}\|\bar{g}_t\|^2)^{2/3} + 80B^{3/2}(1 + \sigma^2 T + \mathbf{E}\sum_{t=1}^{T}\|\bar{g}_t\|^2)^{1/3}$$

This immediately implies that,

$$\mathbf{E}\sum_{t=1}^{T}\|\bar{g}_t\|^2 \le O(1 + L^3 + B^{9/4} + B^{3/2}\sigma^{2/3}T^{1/3}) \tag{31}$$

**Concluding** From Equations (29), (32) it follows that,

$$\mathbf{E}\sum_{t=1}^{T}\|\bar{g}_t\|^2 \le O(M^2 + \kappa^2\sigma^{2/3}T^{1/3}) \tag{32}$$

where $\kappa^2 := B^{3/2} + L^3$, and $M^2 := 1 + L^{9/2} + B^{9/4} + G^{10} + (L^2 G^8)^{3/2}$.

Using the definition if $\bar{x}_T$ together with Jensen's inequality gives,

$$\mathbf{E}\|\nabla f(\bar{x}_T)\| = O\left(\frac{M}{\sqrt{T}} + \frac{\kappa \sigma^{1/3}}{T^{1/3}}\right).$$

which concludes the proof.

### A.3 Additional Proofs

#### A.3.1 Proof of Lemma 6

*Proof of Lemma 6.* Lets define,

$$N_0 = \min\left\{ i \in [n] : \sum_{j=1}^{i-1} a_j \geq a_{\max} \right\} .$$

Thus, we can decompose the sum as follows,

$$
\begin{aligned}
\sum_{i=1}^n \frac{a_i}{(1 + \sum_{j=1}^{i-1} a_j)^{4/3}} &= \sum_{i=1}^{N_0-1} \frac{a_i}{(1 + \sum_{j=1}^{i-1} a_j)^{4/3}} + \sum_{i=N_0}^n \frac{a_i}{(1 + \sum_{j=1}^{i-1} a_j)^{4/3}} \\
&\leq \sum_{i=1}^{N_0-1} a_i + \sum_{i=N_0}^n \frac{a_i}{(1 + \sum_{j=1}^{N_0-1} a_j + \sum_{j=N_0}^{i-1} a_i)^{4/3}} \\
&\leq 2a_{\max} + \sum_{i=N_0}^n \frac{a_i}{(1 + a_{\max} + \sum_{j=N_0}^{i-1} a_i)^{4/3}} \\
&\leq 2a_{\max} + \sum_{i=N_0}^n \frac{a_i}{(1 + a_i + \sum_{j=N_0}^{i-1} a_i)^{4/3}} \\
&\leq 2a_{\max} + 12
\end{aligned}
$$

where the second and third lines use the definition of $N_0$ and definition of $a_{\max}$, the fourth line uses $a_i \leq a_{\max}$, and the last line uses the following helper lemma that we prove in Section A.3.2.

**Lemma 9.** *For any non-negative real numbers $a_1, \dots, a_n \in [0, a_{\max}]$,*

$$\sum_{i=1}^n \frac{a_i}{(1 + \sum_{j=1}^i a_j)^{4/3}} \leq 12 .$$

$\square$

#### A.3.2 Proof of Lemma 9

*Proof of Lemma 9.* Define,

$$N_0 = \max\left\{ i \in [n] : \sum_{j=1}^i a_j \leq 2 \right\} .$$

as well as for any $k \geq 1$

$$N_k = \max\left\{ i \in [n] : 2^k < \sum_{j=1}^i a_j \leq 2^{k+1} \right\} .$$

Now lets split the sum according to the $N_k$'s

$$\sum_{i=1}^{n} \frac{a_i}{(1 + \sum_{j=1}^{i} a_j)^{4/3}} = \sum_{i=1}^{N_0} \frac{a_i}{(1 + \sum_{j=1}^{i} a_j)^{4/3}} + \sum_{k=1}^{\infty} \sum_{i=N_{k-1}+1}^{N_k} \frac{a_i}{(1 + \sum_{j=1}^{i} a_j)^{4/3}}$$

$$\leq \sum_{i=1}^{N_0} a_i + \sum_{k=1}^{\infty} \frac{1}{(2^k)^{4/3}} \sum_{i=1}^{N_k} a_i$$

$$\leq 2 + \sum_{k=1}^{\infty} \frac{2^{k+1}}{(2^k)^{4/3}}$$

$$= 2 + \sum_{k=1}^{\infty} \frac{2^{k+1}}{(2^k)^{4/3}}$$

$$= 2 + 2 \sum_{k=1}^{\infty} \left( \frac{1}{2^{1/3}} \right)^k$$

$$\leq 2 + 2 \cdot \frac{1}{1 - 2^{-1/3}}$$

$$\leq 12 .$$

$\square$

### A.3.3 Proof of Lemma 7

*Proof.* The lemma has two parts.

**Proof of first part.** Recalling that $\tau^* = \max\{t \in [T] : a_t \geq \beta\}$ for $\beta = \min\{1, 1/G^4\}$ implies that $1/a_t \leq 1/\beta \; ; \forall t \leq \tau^*$. Moreover, using the definition of $a_t$ and boundedness of gradients we obtain,

$$(1/a_{\tau^*+1})^{3/2} = (1/a_{\tau^*})^{3/2} + \|g_{\tau^*}\|^2 \leq \frac{1}{\beta^{3/2}} + G^2$$

Defining $\frac{1}{\tilde{\beta}} := \left( \frac{1}{\beta^{3/2}} + G^2 \right)^{2/3}$ implies that,

$$1/a_t \leq 1/\tilde{\beta} ; \qquad \forall t \leq \tau^* + 1 .$$

**Proof of second part.** First note that the function $H(y) := y^{2/3}$ is concave over $\mathbb{R}_+$. Applying the gradient inequality for concave functions imply that,

$$\forall y_1, y_2 \geq 0 \; ; H(y_2) - H(y_1) \leq \nabla H(y_1)^\top (y_2 - y_1) = \frac{2}{3} \frac{1}{y_1^{1/3}} \cdot (y_2 - y_1) .$$

Therefore, for any $t \geq \tau^* + 1$

$$\frac{1}{a_{t+1}} - \frac{1}{a_t} = (1 + \sum_{i=1}^{t-1} \|g_i\|^2 + \|g_t\|^2)^{2/3} - (1 + \sum_{i=1}^{t-1} \|g_i\|^2)^{2/3}$$

$$\leq \frac{2}{3} \frac{\|g_t\|^2}{(1 + \sum_{i=1}^{t-1} \|g_i\|^2)^{1/3}}$$

$$= \frac{2}{3} \sqrt{a_t} \|g_t\|^2$$

$$\leq \frac{2}{3} \sqrt{\beta} G^2$$

$$\leq \frac{2}{3} .$$

where the fourth line uses the definition of $\tau^*$, and the last line uses the definition of $\beta$.

$\square$

### A.3.4 Proof of Lemma 8

*Proof.* Let $b_1, ..., b_n \in (0, b]$ be a sequence of non-negative real numbers for some positive real number $b$, $b_0 > 0$ and $p \in (0, 1)$ a rational number. Then,

$$\sum_{i=1}^{n} \frac{b_i}{\left(b_0 + \sum_{j=1}^{i-1} b_j\right)^p} \leq \frac{b}{(b_0)^p} + \frac{2}{1-p} \left(b_0 + \sum_{i=1}^{n} b_i\right)^{1-p}$$

The proof of this lemma relies on the arguments of Lemma A.1 from [Bach and Levy, 2019] and makes use of Lemma 3 we proved earlier. We consider two cases for the proof depending on whether $b_0 \leq b$ or $b_0 \geq b$.

**Case 1 :** $b_0 \geq b$.

$$\sum_{i=1}^{n} \frac{b_i}{\left(b_0 + \sum_{j=1}^{i-1} b_j\right)^p} \leq \sum_{i=1}^{n} \frac{b_i}{\left(b + \sum_{j=1}^{i-1} b_j\right)^p}$$

$$\leq \sum_{i=1}^{n} \frac{b_i}{\left(\sum_{j=1}^{i} b_j\right)^p}$$

$$\leq \frac{1}{1-p} \left(\sum_{i=1}^{n} b_i\right)^{1-p}$$

$$\leq \frac{b}{(b_0)^p} + \frac{2}{1-p} \left(b_0 + \sum_{i=1}^{n} b_i\right)^{1-p}$$

**Case 2 :** $b_0 \leq b$.
Let us denote a time variable

$$T_0 = \min\left\{i \in [n] : \sum_{j=1}^{i-1} b_j \geq b\right\}$$

Then, we could separate the summation as

$$\sum_{i=1}^{n} \frac{b_n}{(b_0 + \sum_{j=1}^{i-1} b_j)^p} = \sum_{i=1}^{T_0-1} \frac{b_n}{(b_0 + \sum_{j=1}^{i-1} b_j)^p} + \sum_{i=T_0}^{n} \frac{b_n}{(b_0 + \sum_{j=1}^{i-1} b_j)^p}$$

$$\leq \frac{1}{(b_0)^p} \sum_{i=1}^{T_0-1} b_n + \sum_{i=T_0}^{n} \frac{b_n}{(\frac{1}{2}\sum_{j=1}^{i-1} b_j + \frac{1}{2}\sum_{j=1}^{i-1} b_j)^p}$$

$$\leq \frac{b}{(b_0)^p} + \sum_{i=T_0}^{n} \frac{b_n}{(\frac{1}{2}b + \frac{1}{2}\sum_{j=1}^{i-1} b_j)^p} \qquad \text{(Use definition of } T_0\text{)}$$

$$\leq \frac{b}{(b_0)^p} + 2\sum_{i=T_0}^{n} \frac{b_n}{(\sum_{j=1}^{i} b_j)^p} \qquad \text{(Use } b_i \leq b\text{)}$$

$$\leq \frac{b}{(b_0)^p} + \frac{2}{1-p} \left(\sum_{i=T_0}^{n} b_i\right)^{1-p} \qquad \text{(Use Lemma 3)}$$

$$\leq \frac{b}{(b_0)^p} + \frac{2}{1-p} \left(b_0 + \sum_{i=1}^{n} b_i\right)^{1-p}$$

$\square$

## A.4 Numerical Results

In this section we provide numerical performance of STORM$^+$ for a multi-class classification task. Specifically, we train ResNet34 architecture on CIFAR10 dataset using SGD with momentum, STORM and STORM$^+$ , as well as AdaGrad and Adam. We implemented the whole setup in *pytorch* Paszke et al. [2019] retrieving the model and the dataset from *torchvision* package. We executed the experiments on NVIDIA DGX infrastructure. Specifically, our code ran on NVIDIA A100-SXM4-40GB graphics card. We use mini-batches of 100 samples both for training and testing, whiling using the default train/test data split provided in the package.

To be fair to all methods, we fixed all the parameters to their default value except for the learning rate. Then, we executed an initial learning rate sweep over the same logarithmic range for all the algorithms. All methods use a constant learning rate schedule without any heuristic strategies. All methods are run with the best performing initial learning rate after tuning and the results for a single run are presented in Figure 1. In the plots, epoch refers to the number of passes over dataset, *not* number of gradient calls. Per iteration cost of STORM and STORM$^+$ are twice that of other methods with respect to forward/backward passes.

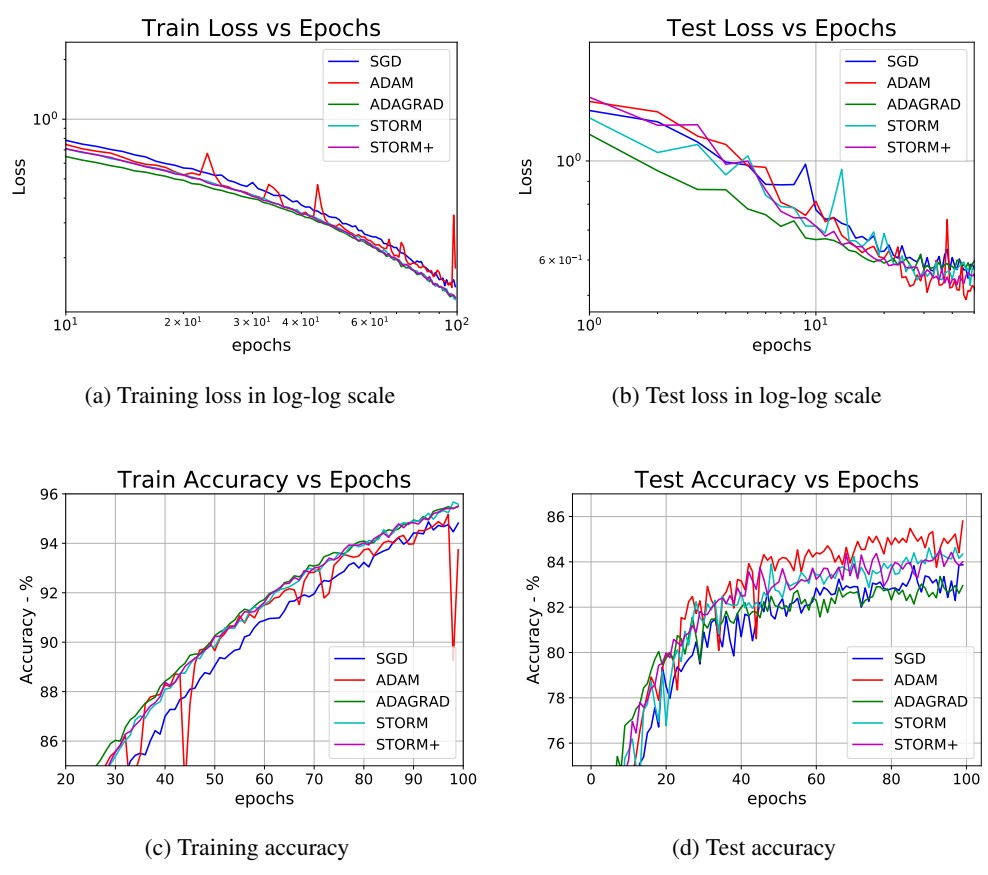

(a) Training loss in log-log scale

(b) Test loss in log-log scale

(c) Training accuracy

(d) Test accuracy

Figure 1: Comparison of SGD and adaptive methods, Resnet34 on CIFAR10

The results do not exhibit a noticeable practical advantage for STORM$^+$ , however, they verify that it achieves comparable performance with respect to other adaptive methods. The performance of STORM and STORM$^+$ are quite close to each other under all 4 metrics. In the training phase, STORM and STORM$^+$ seem to outperform other methods by a small margin, both in training accuracy and training loss. Adam and SGD seem to achieve a relatively small training accuracy and relatively large training loss compared to other methods. In the test phase, we observe a different picture where Adam generalizes slightly better than other methods, followed by STORM and STORM$^+$ as we could see in Figure 1d.

In terms of ease of tuning, provably, $\mathrm{STORM}^+$ does not require the knowledge of *any* problem parameters to operate and only initial step-size tuning suffices, while STORM additionally needs to tune the initial momentum parameter as, in theory, it requires the knowledge of smoothness and bound on the gradients. Adam would need tuning for its moving average parameters $\beta_1$ and $\beta_2$, while SGD has a momentum parameter which is subject to a search over admissible values. Similar to $\mathrm{STORM}^+$ , AdaGrad does not require tuning beyond initial learning rate.