# OpenReview forum: "STORM+: Fully Adaptive SGD with Recursive Momentum for Nonconvex Optimization"
_NeurIPS.cc/2021/Conference — NeurIPS 2021 Poster_

### Official Review · Reviewer_Ft5i · 2021-07-12

**Rating:** 5
**Confidence:** 4

**Summary:**

The paper proposes STORM+, which adopts the STORM method with a new way to set the learning rate (parameter free momentum based). The method also obtains the state-of-the-art convergence result for finding an approximate stationary point.

**Ethics Review Area:**

["I don’t know"]

**Main Review:**

Comments:

1) The idea of applying the adaptive learning rate to the existing method is not new. This could be considered as a marginal work. However, I appreciate the convergence analysis of the paper.

2) Numerical experiments (in the supplementary document) do not show any advantages of STORM+ over STORM. It is true that STORM+ does not require to tune the parameters because of the parameter-free choices of parameters, but from the experiments, it is not clear whether STORM uses the best tuning parameters.

3) Comparing to other methods, STORM+ also does not show the benefits. This would raise a question why people have to consider this method but not using Adam. I would suggest the authors provide more experiments to verify the effectiveness of the proposed method.

4) Since the paper focuses on the practical side of STORM, it would be nice to bring the numerical experimental part into main content.

5) Hybrid-SGD in [Tran-Dinh et al, 2019] does not require “bounded values” and “bounded gradients” assumptions. Could you please have some discussion of why the proposed method has more advantage than Hybrid-SGD? I do not see much comparisons with this method.



**Time Spent Reviewing:**

5

---

> ### Author Response · Authors · 2021-08-10
> **Author Response**
>
> Thank you very much for your evaluation of our work! We respond to each of your comments below:
>
> **“The idea of applying the adaptive learning rate to the existing method is not new. This could be considered as a marginal work. However, I appreciate the convergence analysis of the paper.”**
>
> We respectfully disagree. The development of adaptive methods has had a huge impact on training ML models, and methods like AdaGrad and Adam play a central role in the ability to effectively train large-scale models.
> For instance, AdaGrad is essentially the adaptive version of Zinkevich’s Online Gradient Descent or SGD, and originally developed for online convex optimization. However, it was later discovered to have many favorable properties in large-scale learning problems such as its practical performance in **training** neural networks. Moreover, algorithmic construction and theoretical contributions of AdaGrad paper have paved the way for many adaptive optimization methods that we know of today.
>
> In STORM+ we describe a novel way to **jointly and adaptively** set the learning rate and momentum parameter in a parameter free way that also obtains optimal convergence rates.
> We believe that in the future this may inspire new heuristics to training large scale models that may improve on the current state-of-the-art.
>
>
> **Numerical experiments and benefits.**
>
> In our numerical experiments, for the sake of fairness, we only tune learning rate for all the methods. Regarding the comparison between STORM and STORM+, STORM additionally needs to tune the momentum parameter. Since $a_t$ is a monotonically non-increasing quantity, tuning means setting the initial value, i.e., its upper bound. As it gets smaller, the gradient estimator $d_t$ becomes more biased and it has a negative effect in convergence properties of the algorithm. We tried values smaller than 1 but it didn’t provide any considerable advantage to STORM. We will provide a detailed comparison in the final version.
>
> In fact, our work focuses on the theoretical side and the experimental study only serves as a proof of concept that our approach actually works. Again, we believe that in the future our work may inspire new heuristics to training large scale models, as well as new proof techniques for parameter-free algorithms.
>
>
> **Comparison to Hybrid-SGD in [Tran-Dinh et al, 2019]**
>
> To make it clear once again, our main objective is obtaining optimal convergence rates in the ``expectation over smooth, nonconvex losses’’ setting with a **parameter-free algorithm** that runs without the knowledge of smoothness, gradient bound or variance bound. While doing so, we propose a new way to concurrently adapt the learning rate and momentum parameters as well as a novel analysis technique to handle adaptive parameters. We refer to Hybrid-SGD in our related work, but we will make our discussion on it more detailed.
>
> - First, Hybrid-SGD is a non-adaptive method, which doesn’t guarantee convergence without knowing Lipschitz constant or gradient variance.
>
> - Second, the single-loop variant (most relevant to ours) makes 3 oracle calls per iteration while ours does 2 and it computes an initial estimator with a batch size of order $T^{1/3}$. Additionally, it requires the knowledge of variance, which is again not available in practice.
>
> - Lastly, the bound of Hybrid-SGD is only meaningful once the error of the first iterate compared to the optimal solution is bounded.

---

> > ### Comment · Reviewer_Ft5i · 2021-08-29
> > **Response**
> >
> > I thank the authors for your response.
> >
> > We had some discussion about your results. Unfortunately, I still keep my score unchanged because in my opinion I think the novelty of the paper is limited. I would suggest the authors to submit your work to some optimization journal. I think the quality of this paper is more suitable for a journal publication. Numerical experiments also need to be improved.

---

### Official Review · Reviewer_CNhs · 2021-07-13

**Rating:** 6
**Confidence:** 4

**Summary:**

In this paper, the authors provide a parameter-free version named STORM+ of the STORM algorithm. STORM+ uses the same single-loop recursive momentum (which leads to implicit variance reduction effects) and constant batchsizes as in STORM, but features a parameter-free design by constructing the learning rate \eta_t and momentum coefficient a_t using past parameters and momentums (as a comparison, STORM needs to tune these two factors manually). Theoretically, the authors show that STORM+ achieves the same near-optimal complexity as STORM but without needing to tune the parameters to ensure this guarantee.


**Limitations And Societal Impact:**

Yes

**Main Review:**

After response:

Thanks a lot for your response. I appreciate the novel analysis in this paper to handle the hyperparameters' adaptivity in the original STORM method, and after my second reading, I feel that I am too picky at the assumptions and experiments in this paper. Therefore, I increase my score to 6.
....................................................................................................................................................................................

In this paper, the authors provide a parameter-free version named STORM+ of the STORM algorithm. STORM+ uses the same single-loop recursive momentum (which leads to implicit variance reduction effects) and constant batchsizes as in STORM, but features a parameter-free design by constructing the learning rate \eta_t and momentum coefficient a_t using past parameters and momentums (as a comparison, STORM needs to tune these two factors manually). Theoretically, the authors show that STORM+ achieves the same near-optimal complexity as STORM but without needing to tune the parameters to ensure this guarantee. My detailed comments are given as follows.

Pros:
1.	This paper is easy to following and overall well-written. The motivation is very clear, i.e., replacing the manually tuned learning rate and the momentum coefficients in STORM with adaptive ones based on past iterate information. The proof idea is easy to get based on the sketch in Section 5. Basically, it seems to me that the authors try to add d_i into \eta_t and a_t so that the resulting convergence results also contain a summation of d_i, and such a summation can be well handled after do the telescoping for the final convergence analysis. Overall, the idea is reasonable to me.

2.	It is good to show that the resulting scheme can achieve the same optimal complexity as STORM. Parameter-free optimization and variance reduction are both of great interest to the community, and of course parameter-free algorithm with near-optimal performance guarantee should be of interest.

Cons:
1.	However, I have several concerns on the assumptions and the empirical justification. First, I note that compared to the analysis for STORM, this paper seems to additionally assume that the function value of f is bounded, i.e., max_{x,y\in R^d} |f(x)-f(y)|<\infty. As a result, the comparison to STORM in this paper here is not that fair. In addition, this is a very strong assumption as x is allowed to be any vector in R^d, which can be violated, e.g., for simple linear function f(x)=wx. I am wondering whether such value boundedness is necessary for the adaptivity proof. In STORM paper, they assume the gradient is bounded (as also made in this paper) because they can use this assumption to remove the variance factor \sigma in the learning rate. I am thinking whether this bounded value assumption is further used  to remove other manually tuned parameters? I hope the authors can explain a little bit.

2.	No experiments are provided in this paper. Parameter-free optimization is interesting because it will help to eliminate the expensive hyperparameter selection process while achieving the same promising performance. STORM not only achieves the optimal complexity, but also has good practical performance (comparable to Adam). I strongly encourage the authors can test the performance of STORM+ and figure out whether it is comparable to (even better if it can outperform) STORM with the best hyperparameter tuning. I wish to see such a comparison in the authors’ response.

Overall, I feel this paper contains some interesting designs and analysis, but given my concerns in the assumption and the lacking experiments, I cannot accept it currently. However, I am open to increase my score based on other reviewers’ comments and the authors’ response.


**Time Spent Reviewing:**

6

---

> ### Author Response · Authors · 2021-08-10
> **Author Response**
>
> Thank you very much for your constructive comments and detailed explanations!
>
> To begin with, your intuition about the analysis and algorithmic design is very precise. Indeed, due to the coupling between gradient estimates $d_t$ and learning rate $\eta_t$, we accumulate $d_t$ in the learning rate as opposed to STORM, which accumulates gradients $g_t$.
>
> **Assumption regarding bounded values**
>
> Please note that in the original STORM paper as well as papers analyzing SGD in the non-convex setting, it is assumed that $(f(x_1) - f(x*)$ is bounded. Essentially, our analysis only requires that $f(x_t) – f(x*)$ is bounded for all $t \in \{ 1,2, …, T \}$, we will make this clear in the main text and update the definition of this assumption accordingly. This approach helps us trade-off the complicated parameter choice of STORM that requires the knowledge of $L$ and $G$, which are impossible to set in practice.
>
> Moreover, the assumption of bounded function values holds in several interesting scenarios. To exemplify, for neural networks with sigmoid activation functions this assumption should hold. Actually, it is **enough that the activations of the output layer are bounded (e.g. sigmoid activations)** to ensure that the function values are bounded. Another example is robust regression, where oftentimes one uses a robust bounded loss rather than the square loss in order to better handle outliers. One such popular robust loss function is the Welsch loss (see e.g. this paper [1] ) which is bounded. Note that using the Welsch loss instead of the square (L2) loss leads to a robust, stochastic, non-convex problem, which is extremely relevant to our work.
>
>
> **Linear function example  $f(x)=wx$**
>
> Please note that your example with the linear function is not suitable since in this case the gradient is fixed everywhere and there does not exist a stationary point.
>
>
> **Experiments**
>
> In the supplementary material, in Appendix A4, we do provide experimental study showing that our method has comparable performance to the original STORM paper . Our observations are in parallel with that of the original STORM paper’s; training performance is slightly better than other methods, while Adam is slightly better in the test phase.
>
>
> **References:**
>
> [1] J. T. Barron, "A General and Adaptive Robust Loss Function," 2019 IEEE/CVF Conference on Computer Vision and Pattern Recognition (CVPR), 2019, pp. 4326-4334, doi: 10.1109/CVPR.2019.00446.

---

> > ### Comment · Reviewer_CNhs · 2021-08-29
> > **Thanks for the response**
> >
> > Dear authors,
> >
> > Thanks a lot for your response. I appreciate the novel analysis in this paper to handle the hyperparameters' adaptivity in the original STORM method, and after my second reading, I feel that I am too picky at the assumptions and experiments in this paper. Therefore, I increase my score to 6 toward a weakly acceptance, and with more comprehensive experiments, I feel this work deserves a 7.
> >
> > Best,
> > Reviewer

---

### Official Review · Reviewer_m4aE · 2021-07-16

**Rating:** 6
**Confidence:** 5

**Summary:**

This paper is interesting and technically sound. It devises a thoroughly parameter-free algorithm named STORM+, on the basis of STORM. STORM+ obtains the optimal theoretical results and achieves comparable performance with respect to other adaptive methods. The merit of this paper is that it does not require the knowledge of any problem parameters and only initial step-size tuning suffices.

**Limitations And Societal Impact:**

More experiments should be conducted to demonstrate the effectiveness of the proposed method.

**Main Review:**

1. Perhaps several important findings/conclusions can be better presented in the form of theorems or lemmas, e.g., contents in Line 141-142 and Line 156-159.

2. The meaning of several sentences is not clear (e.g., in Line 209). Careful proofreading is recommended.

3. STORM+ and its simplified ones achieve the optimal rate $\frac{1}{\epsilon^3}$. What are the main differences between them? More discussion is needed to make this paper more solid.

4. Detailed derivations of $M_t$ in Line 413 are preferable, as well as $Y_t$ in Line436.

5. All mathematical symbols or operators used in the text are required to be defined.

6. More discussion and comparison between STORM+ and the existing nonconvex stochastic optimization methods are recommended.

7. The reviewer believes that the main conclusions of the article are solid. However, the current version is not suitable for publication and the reviewer would like to suggest the authors to make serious revision to the article in terms of logic, presentation and organization.

Typos:

Line 146: $...\leq -\frac{1}{\eta_T}\Delta_{t+1} + …$ $\rightarrow$ $...\leq -\frac{1}{\eta_T}\Delta_{T+1}+ …$

Line 204: he$\rightarrow$the


**Time Spent Reviewing:**

10h

---

> ### Author Response · Authors · 2021-08-10
> **Author Response**
>
> Thank you for your constructive comments. Please find our responses to your comments below.
>
> **Presenting the results in sections 5.1 and 5.2 in separate Theorems**
>
> Agreed. We will present our result in section 5.1 and 5.2 in separate Theorems.
>
>
> **Difference between STORM+ and simplified STORM+**
>
> Note that we explicitly state the difference between them in Section 5.2 lines 160-162.
> Concretely, the main difference between STORM+ and its simplified version is that simplified STORM+ uses a deterministic momentum parameters, i.e., $a_{t+1} = 1/t^{2/3}$, while original STORM+ uses $a_{t+1} = 1 / (1 + \sum_{i=1}^{t} \| g_i \|^2)^{2/3}$, which is data dependent.
> For that reason STORM+ adapts to the variance while simplified STORM+ does not.
>
>
> **We include the analysis of simplified STORM+ for two reasons:**
>
> (i) To convey the main ideas and build the intuition towards the more involved analysis of the STORM+ algorithm.
> (ii) To demonstrate that adaptivity of both learning rate and momentum is essential in order to obtain full adaptivity.
>
>
> **Presentation and organization**
>
> Thank you for your other minor comments regarding presentation and organization.
> Incorporating your valuable feedback requires a very minor revision of the paper. And we believe that with this it will be ready for publication.

---

> > ### Comment · Reviewer_m4aE · 2021-08-29
> > **Thanks for the response**
> >
> > I appreciate that the submission gave a completely parameter-free adaptive algorithm, which is crucial in the practical application of the algorithm. However, the experimental results do not seem to confirm this well. Therefore I am going to lower my rating to 6.

---

### Official Review · Reviewer_ZL4A · 2021-07-18

**Rating:** 5
**Confidence:** 4

**Summary:**

This paper proposes an adaptive variant of the recently proposed STORM algorithm for minimizing expectations of smooth, nonconvex losses. By doing so, it removes the hyperparameters necessary for running STORM.
It offers an analysis of the method and rates of convergence, but no empirical validation of the method.

**Main Review:**

The main argument for this method is practical : STORM requires parameters that are hard to access, namely the common lipschitz constant of the functions appearing in the expectation. It seems to me that it is therefore crucial to show experiments validating the proposed adaptive method. Additionally, it would be beneficial to compare the adaptive variant to the sublinear momentum variant on typical problems.
This explains my rating. I will be glad to modify it given empirical evidence of the proposed algorithm's performance. I would recommend a minima reproducing the experiments from the initial STORM paper, adding the STORM+ variants considered in this paper.

The worst case bound adapts to the magnitude of noise: is this the case in practice on toy experiments? It would be nice to check whether this is an artifact of the proof/worst case, or if this is also true in practice. Such experiments would help me modify my rating.

What is the point of section 5.2 ? I would argue to move this to appendix, saying that a version of the algorithm w/ sublinear momentum parameter also achieves the same aymptotic convergence rate, although the non-asymptotic rate does not adapt to the variance of the stochastic gradient estimator.
Rather than provide the analysis for the case sigma=0 and the case w/ sublinear momentum parameter, I would also rather see an extended sketch of the analysis of STORM+ in the main paper. It would also be helpful for the reader to re-state things when they are needed.

The paper is not very well written, with quite a few typos. I would recommend working on the flow of the paper and its presentation : re-stating asumptions/lemma results when needed, relying less on bullet-point style presentation (cf line 182 - 206).

Comments on form:
Typo line 30: hanging sentence
Line 82: forgot the bar on x
It should be “Such that” instead of “such” in all the conditions (bounded values etc)
Line 84: why is x* a global minima, rather than a stationary point?
Line 92: notation consistency: maybe use greek letters for scalars? (a_t -> rho_t or beta_t to be consistent with the literature?)

Lemma 2: why is it written that p must be rational? This isn’t used in the proof in the appendix.
Proof of lemma 2 : the induction hypothesis must hold for n-1 >= 1, not > 1. It also needs to be said somewhere that b1 cannot be 0.

**Time Spent Reviewing:**

2

---

> ### Author Response · Authors · 2021-08-10
> **Author Response**
>
> We would like to thank the reviewer for their comments and constructive approach in their evaluation.
>
> **“The main argument for this method is practical”**
>
> While the motivation is practical, our submission focuses on the theoretical question on whether one can design a fully adaptive and optimal method for finding a stationary point in the non-convex setting. An important implication of our work is that this indeed can be done by simultaneously adapting the learning rate and momentum parameter in a dependent and novel manner.
> Note that theoretically grounded adaptive methods for convex problems had a huge impact on the ML field (e.g. AdaGrad and Adam). And we hope that our new theoretical findings will lead to new heuristics that may impact the way training is done nowadays.
>
>
> **“I will be glad to modify it given empirical evidence of the proposed algorithm's performance. I would recommend a minima reproducing the experiments from the initial STORM paper, adding the STORM+ variants considered in this paper.”**
>
> Actually, in the supplementary material, in Appendix A4, we do provide experimental study showing that our method has comparable performance to the original STORM paper. Our observations are in parallel with that of the original STORM paper’s; training performance is slightly better than other methods, while Adam is slightly better in the test phase.
>
>
> **Experiments showing STORM+  adapts to noise magnitude:**
>
> We have cooked up such an example using Welsch loss, a smooth, non-convex loss function. We run STORM+ with deterministic gradients, and stochastic gradients with two different variance values. Stochastic gradients are generated by injecting zero-mean noise to full gradient. We execute the stochastic methods 20 times and average the plots over those runs.
>
> *Please find the plot [here](https://imgur.com/a/mmymydd)*
>
>
> **“What is the point of section 5.2 ?” (Analysis of simplified STORM+)**
>
> We have included section 5.1 (analysis of noiseless case), and section 5.2 (simplified STORM+ analysis) to convey the main ideas of the more complicated STORM+ analysis. We believe that this substantially improves the readability and presentation in the paper.
> As one can see from section 5.3 and from the appendix, the analysis of STORM+ is much more involved and technical.
> Another reason we include section 5.2 on simplified STORM+ is to demonstrate that we cannot adapt to variance with a simple data-independent momentum term. Therefore, we wanted to emphasize that the intricate relationship between data-dependent $\eta_t$ and data-dependent $a_t$ is essential to achieve the rate interpolation between $1/\sqrt{T}$ and $1/T^{1/3}$.
>
>
> **Line 84: why is $x^\star$ a global minima, rather than a stationary point?**
>
> This definition is standard in works that analyze first order methods for finding stationary points in non-convex problems, including the original STORM paper.  Technically, it is required to ensure the positivity of the differences $f(x_t) - f(x*)$, it is guaranteed to be positive if and only if $x*$ is a global minimum.
>
>
> **“Lemma 2: why is it written that p must be rational?”**
>
> Indeed p does not have to be rational, and the proof holds for any real number $p \in (0, 1)$. We will correct it.
>
>
> We thank the reviewer for pointing out some typos and minor comments. We will correct them in the final version. Also, we will list our assumptions with better referencing and correctly refer to them in the Theorem statements.

---

> ### Author Response · Authors · 2021-08-25
> **We would like to follow up on our response**
>
> We hope that our response to your concerns were informative and provided you with the explanations you were seeking for.
>
> We tried to answer your remarks in the clearest way possible and also provided a toy example plot that validates our rate interpolation as you requested. If you have found our response satisfying and convincing, we would appreciate it greatly if you could reconsider your evaluation and scoring. If you have any further remarks/concerns, please reach out to us! We will do our best to clarify them, and would be happy to engage in a constructive discussion with you (and with other reviewers, as well).

---

> > ### Comment · Reviewer_ZL4A · 2021-08-30
> > **Response**
> >
> > Thank you for your answer, and for your work! I appreciate your clarifications to my and the other reviewers' comments.
> >
> > *On empirical results*
> >
> > The results obtained show that the methods achieves similar levels of accuracy as other, well known methods. One caveat here. Having parameters may actually benefit other methods here: tuning their parameters leaves the hope of achieving better results. For STORM+, these results are pretty much final, and no improvements can be expected, which means in practice the method would not be used. Showcasing at least one practical setting giving a clear advantage to STORM+ would greatly benefit the method and its dissemination.
> >
> > Obtaining an adaptive version of STORM is of theoretical interest, and I appreciate this work for this. On the other hand, I'm not sure that the current iteration of the paper will yield the interest/dissemination that this result would deserve. As other reviewers have argued, the paper needs a deeper rewrite, reorganizing proofs. The current empirical results are also not very satisfying for a method with no degrees of freedom left.
> >
> > I'm willing to increase my score to 5, but still do not recommend acceptance for now.

---

> > > ### Author Response · Authors · 2021-09-01
> > > **Thank you for following up on our response**
> > >
> > > Thank you for reconsidering your evaluation and scoring! However, we recognised that there is fundamental misconception regarding our work, which we would like to clarify further.
> > >
> > > "Parameter-free" does not mean that the algorithm has no "tunable" parameters/step-size. It means that one can set the step-size and momentum parameter using a dynamic rule which **does not require the knowledge of any problem parameters** including smoothness constant, variance and gradient bounds.
> > >
> > > Regarding your statement about parameter tuning, one could still tune our algorithm. Recall that we set the numerator of $a_{t+1}$ and $\eta_t$ as $1$. It is in fact a placeholder for the initial value of those parameters which we can essentially tune by setting it to a smaller/larger number in practice. Theoretically, it could be optimised to yield better constants in the convergence bound. This is equivalent to tuning in practice.
> > >
> > > While original STORM and other methods require complicated rules to set the algorithmic parameters, our rule only relies on observed data. For instance, STORM must know the Lipschitz constant $L$ and a uniform upper bound $G$ on gradient norms in order to guarantee convergence. In practice, it is very unlikely, if not impossible, to know/access these values. Especially with very large datasets and in streaming/online data regime, it is almost impossible to compute them. We theoretically prove a more "realistic" algorithmic framework with an easy parameter setting strategy, which works in any of those cases.
> > >
> > > With that perspective, our method is much more suitable for practical usage as it doesn't need problem parameters that aren't viable to compute. We show that our method also has comparable, if not better, performance than most methods we compare against. Our findings are in parallel with STORM paper. In a way, we provide theoretical verification and a rather technical perspective to the process of tuning, closing the gap between theory and practice.

---

### Decision · Program_Chairs · 2021-09-27

**Decision:**

Accept (Poster)

**Comment:**

This paper generated a lot of discussion among reviewers. Overall it seems that the contribution is novel and provides an interesting new learning rate schedule meriting acceptance. However, the claims of optimality in the rates are not well-supported. The extra assumption that the objective is bounde above and below may invalidate current lower bounds.